A systematic review of fluralaner as a treatment for ectoparasitic infections in mammalian species

Jiang Yuanting
http://orcid.org/0000-0002-2754-7757 Old Julie M. j.old@westernsydney.edu.au
School of Science, Western Sydney University , Penrith, NSW , Australia
Beddoe Travis
Electronic publication date: 2025 Mar 12
Publication date: 2025
Volume: 13
Electronic Location ID: e18882
Received 2024 Sep 30; Accepted 2024 Dec 28
Copyright: © 2025 Jiang and Old
Copyright year: 2025
Copyright holder: Jiang and Old
License: This is an open access article distributed under the terms of the Creative Commons Attribution License, which permits unrestricted use, distribution, reproduction and adaptation in any medium and for any purpose provided that it is properly attributed. For attribution, the original author(s), title, publication source (PeerJ) and either DOI or URL of the article must be cited.
License URL: https://creativecommons.org/licenses/by/4.0/

Keywords: Bravecto®, Exzolt, Mammal, Efficacy, Ectoparasite, Treatment, Side effect, Pharmacokinetic, Safety, Environment

Funding: The authors received no funding for this work.

==============================
Fluralaner (Bravecto™) is a novel isoxazoline ectoparasiticide used for controlling ectoparasites of domestic mammals and is increasingly being used on wildlife. The aim of this systematic review was to evaluate the efficacy, pharmacokinetics, and safety of fluralaner on mammals given its increasing use. The search was performed in GoogleScholar and the SciFinder databases using the terms ‘fluralaner’ and ‘Bravecto™’, and was concluded on 30th August, 2024. A total of 250 references were initially saved and reduced to 121 peer-reviewed journal articles using PRISMA guidelines, based on the removal of duplicates, those published in low quality journals (ranked less than Q2), and limiting publications to clinical trials. Articles were then categorised and ranked using the level of evidence, Cochrane ‘risk of bias’ assessment tool, methodological quality, and study size. Overall, the efficacy of fluralaner has been assessed on 14 mammalian species, and pharmacokinetic investigations conducted on 15. Fluralaner was mostly effective at treating some ectoparasites on captive individuals when there was little chance of re-infection; however, it did not prevent bites from blood-sucking ectoparasites and could not prevent blood-borne pathogen transfer to host animals. Very few studies have investigated the pharmacokinetics of fluralaner, and hence were difficult to compare; however, wombats differed greatly from their eutherian counterparts in their Cmax and t½ values and require further investigation. Overall, fluralaner was deemed moderately safe; however, most studies were classified as fair in terms of quality and most studies were based on small or very small sample numbers. Nineteen studies reported side effects, one of which included signs of severe neurological toxicity, with most of the articles not reporting on safety either positively or negatively. Concerns were raised regarding the extended time fluralaner was detected in faeces and subsequently environmental contamination is a concern. No longer-term impacts of the use of fluralaner have been investigated, and wider implications of the use of this ectoparasiticide have not yet been assessed.

Introduction

Ectoparasites infest livestock, domestic and wildlife species worldwide, and are important ecologically as they can regulate population numbers (Tompkins & Begon, 1999). They utilise host tissues (blood, lymph, skin, fur, and feathers) as their food resource, resulting in skin lesions, reduced hair quality and/or loss, their skin may crack and crust, with epidermal and dermal lesions of hyperkeratosis, vasculitis, and aggregates of inflammatory cells forming (Chanie, Negash & Sirak, 2010). In some cases ectoparasites may cause more serious impacts on individuals including changes in metabolic rates and body weights (Khokhlova et al., 2002), and can even cause poisoning, paralysis, respiratory failure, and death (Hall-Mendelin et al., 2011). Moreover, ectoparasites influence behaviour of their hosts (Kluever, Iles & Gese, 2019), and can act as vectors for pathogenic organisms (Dvorak, Shaw & Volf, 2018).

Ectoparasites of livestock, such as lice, lead to reduced production outputs, with farmers forced to spend considerable amounts of money each year to minimise associated production loses (Byford, Craig & Crosby, 1992). Likewise, significant costs are associated with reducing ectoparasites of our beloved cats and dogs, mostly fleas, but also ticks, mites, and lice associated with a range of ‘pocket pets’.

The impacts of ectoparasites on wildlife are also significant, particularly for endangered species, which can lead to reduced health, lowered reproductive output, and death (McCallum, 1994), ultimately posing a conservation risk to species. Sarcoptic scabiei is a global threat to wildlife and has been documented in well over 200 mammalian species (Pence & Ueckermann, 2002) including coyotes (Canis latrans) and bears (Ursinus americanus) in North America (Niedringhaus et al., 2019), Iberian ibex (Capra pyrenaica) in Europe (Sarasa et al., 2010), and infections of Sarcoptes scabiei on wombats in Australia are fatal if not treated (Old et al., 2018).

The need to treat many different species, safely and effectively, using different products and methods is ever present. For free-ranging wildlife species, the treatment and method can be even more problematic (Churgin et al., 2018; Rojas et al., 2024; Wall, 2007). Bravecto™ is a veterinary medicine used for the treatment of fleas, mites and other ectoparasites. Fluralaner (FLU) is the main constituent of Bravecto™, with the content varying depending on treatment method, but is commonly used as an oral or topical application. ExzoltTM is also available as a FLU pour-on treatment for cattle in some countries. FLU was first approved for use in Australia in 2015 for the treatment of dogs (https://apvma.gov.au/node/12911), having been approved overseas (European Medicines Agency, 2023; USFDA, 2023a). As the drug has been further promoted, the scope of application has been gradually expanded from domestic pets to trials on wildlife (Van Wick et al., 2020; Wilkinson et al., 2021)

FLU is a novel isoxazoline ectoparasiticide. It can effectively block the γ-aminobutyric acid-gated chloride and l-glutamate-gated chloride channels (Gassel et al., 2014; Weber & Selzer, 2016). These receptors are found in the central nervous system of invertebrates, generating inhibitory potentials to facilitate the correct integration of neuronal signals, and in peripheral neuromuscular sites, promoting muscle relaxation. Eventually the invertebrate will die because of a central nervous system hyperexcitation or muscle paralysis (Gassel et al., 2014).

Currently there is a lack of comprehensive reviews of FLU’s effectiveness and safety, particularly across different species and via different routes of administration, leading to perceived variations in efficacy. Furthermore, inconsistencies in the reporting of side effects has led to public concerns regarding safety, with some government departments issuing alerts regarding adverse neurological events on cats and dogs (USFDA, 2023b), demonstrated adverse effects on fertility, embryotoxicity and other adverse effects on offspring in rats, and skeletal and visceral malformations in rabbits (Merck, 2024; MSD, 2023). The reasons for the lack of consistent efficacy and safety may be due to differences in administration, physiology or other host factors. We conducted a systematic review to assist in filling the paucity of knowledge, specifically we assessed the effectiveness, and safety, of FLU in different mammalian species using varying administration routes.

Materials and Methods

Search strategy and selection

A systematic review of the effectiveness and outcomes from the use of fluralaner was conducted by the authors (YJ and JO). All articles included in the study selection process were agreed on by the authors and no referees were required. The search terms “fluralaner” and “Bravecto®” were used to identify peer-reviewed journal articles in the GoogleScholar and SciFinder databases. The search was concluded on 30th August, 2024 and had no restrictions placed on the search (include date restrictions), except that they were in the English language. If they were not relevant to the study, they were excluded, such as descriptive chemical studies or those only mentioning fluralaner or Bravecto® in unrelated publications.

The list of 250 references was imported into an Endnote reference library, to organise, remove duplicates, and categorise studies for systematic screening, which resulted in 229 peer-reviewed journal articles. After removing low quality journals (those ranked less than Q2, using Scimago Journal & Country Rank (https://www.scimagojr.com/)), 179 articles remained. The list of journal articles was further reduced to 121 after manually screening only full-text studies involving clinical ectoparasite trials or observations of FLU use on mammals (Fig. 1). Articles removed included a comparative review of treatment efficacy against specific insects, and other reviews including two systematic review articles on generalised demodicosis in canines, ticks, and fleas. The systematic review process is summarised in a Preferred Reporting Items for Systematic Reviews and Meta-analyses (PRISMA) diagram (Fig. 1), showing the final number of articles assessed and included in the review was 121, and agreed on by both authors.

Figure 1 PRISMA diagram.

The figure shows the flow of study identification and selection. The original database search resulted in 250 records from GoogleScholar and included an initial screen based on title and abstract, and only included articles written in English. After duplicates were removed, there were 229 unique citations eligible for further screening. The second screening excluded 102 records for the following reasons: articles not in Q1 or Q2 journals, not involving clinical trials on mammals, not relating to ectoparasite treatments; resulting in 121 records to assess by screening the full-text articles.

Level of evidence and methodological quality

The data from each article was extracted and assessed for level of evidence (LOE) and quality and values, by the two authors, recorded in a Microsoft Excel spreadsheet and verified for consistency. The articles’ level of study design were evaluated according to previous systematic reviews (Schraven, Stannard & Old, 2021). Each article was assigned as a randomised controlled trial (LOE 1), non-randomised controlled trial (LOE 2), or experimental control trial (LOE 3) (Boller & Fletcher, 2012; Goggs et al., 2014). To assess quality, articles were then classified using the Cochrane ‘risk of bias’ assessment tool (Higgins et al., 2019). Briefly, each of the five bias assessments (selection, performance, detection, attribution, and reporting) were given a high (+), low (−), or unclear (?) rating, except for selection bias, where the study was LOE 2 (i.e., designated none, because it is a non-randomised trial), and performance bias, where the study was designated LOE 3, because the blinding of participants or owners was not required (i.e., designated N/A, because it is an experimental control trial). The overall bias was determined from the ratings of the five bias assessments (low, low/moderate, moderate, moderate/high, and high). Bias assessments were determined by addition of each of the bias ratings. For example, where all ratings were negative it was rated as low bias, where two ratings were positive it was rated as moderate bias, and where all ratings were positive it was rated as high. If ratings were unknown (?) or N/A, it either increased or decreased the rating from the overall addition of each rating, respectively.

Further quality assessments involved allocating a quality of enrolment score whereby articles were ranked poor, fair, or good based on Jessen et al. (2015). Articles were given a good ranking when the disease/health status of the target animals were confirmed by a professional (i.e., veterinarian), clinical examinations were performed and blood collections taken for pharmacokinetic studies were reported, and information on the study subjects included age, sex, and weight. Articles ranked as fair had the study subjects’ health/disease status confirmed by a professional (i.e., veterinarian or scientist), had clinical examinations reported but only some information on age, sex, and weight was included. Articles where the health/disease status was not confirmed by a professional, the diagnostics were vague, and little to no information on age, sex, and weight were ranked as poor.

In addition, the size of the study was classified based on previous systematic reviews (Olivry, Mueller & The International Task Force on Canine Atopic Dermatitis, 2003). Groups were classified as large (>50), medium (20–50), small (10–19), or very small (<10).

Results and Discussion

This review encompasses 121 studies from 2014–2024 on FLU’s effectiveness against ectoparasites in 14 mammalian species. Of the 121 articles published from 2014 to 2024 used in the systematic review (Table 1), all were classified as peer-reviewed journal articles, and were published in a total of 28 different, Q1 (n = 113) and Q2 (n = 21), publications. All journals were assessed as Q1 or Q2. Most of the articles were classified as LOE 1 (n = 82), LOE 2 (n = 24), and LOE 3 (n = 28); and all articles were ranked as having low (n = 118), low/moderate (n = 5), or moderate (n = 10) levels of bias, except one which was ranked moderate/high. However, when assessed for further quality based on diagnosis of health/disease status by a professional, levels of clinical examination, blood collection, and physical attributes, many were ranked as poor (n = 13) or fair (n = 98), and only 23 ranked as good. Furthermore, study size of the groups varied with 13 categorised as large, 16 medium, 30 small, and 75 very small (Table 1). One article describing a model resulted in a classification as not applicable in sample quality and size. When multiple trials were reported within an article, the highest level of quality was recorded for article quality.

Table 1 Categorisation of journal articles including level of evidence, quality, size, bias and Q-journal.

Species	Level of evidence and study design	Enrolment quality	Size of study group	Selection bias	Performance bias	Detection bias	Attrition bias	Reporting bias	Overall bias	Q	References	
Atelerix albiventris	LOE 3	Fair	Very small	+	N/A	–	–	–	Low	1	Romero et al. (2017)	
Bos taurus	LOE 1	Fair	Medium	–	–	–	–	–	Low	1	de Aquino et al. (2024)	
Bos taurus	LOE 1	Fair	Small	–	–	–	–	?	Low	1	Gallina et al. (2024)	
Bos taurus	LOE 1	Poor	Large	+	–	–	–	?	Low/moderate	1	da Costa et al. (2023)	
Bos taurus	LOE 1	Poor	Small	?	–	–	–	–	Low	1	Reckziegel et al. (2024)	
Bos taurus	LOE 2	Fair	Medium	None	–	–	–	?	Low	1	Zapa et al. (2024)	
Canis familiaris	LOE 1	Fair	Large	–	–	–	–	+	Low	1	Rohdich, Roepke & Zschiesche (2014)	
Canis familiaris	LOE 1	Fair	Large	–	–	–	–	–	Low	1	Taenzler et al. (2016b)	
Canis familiaris	LOE 1	Fair	Large	–	–	–	+	–	Low	1	Chiummo et al. (2020)	
Canis familiaris	LOE 1	Fair	Large	–	?	–	+	–	Low/moderate	1	Meadows, Guerino & Sun (2014)	
Canis familiaris	LOE 1	Fair	Large	–	+	–	+	–	Moderate	1	Meadows, Guerino & Sun (2017a)	
Canis familiaris	LOE 1	Fair	Medium	–	–	–	+	–	Low	1	dos Santos et al. (2024)	
Canis familiaris	LOE 1	Fair	Medium	–	–	–	–	?	Low	1	Petersen et al. (2020)	
Canis familiaris	LOE 1	Fair	Medium	–	–	–	–	–	Low	1	Taenzler et al. (2014)	
Canis familiaris	LOE 1	Fair	Medium	–	–	–	?	–	Low	1	Wengenmayer et al. (2014)	
Canis familiaris	LOE 1	Fair	Medium	–	–	–	+	–	Low	1	dos Santos et al. (2022)	
Canis familiaris	LOE 1	Fair	Medium	–	–	–	–	–	Low	1	Fisara & Webster (2015)	
Canis familiaris	LOE 1	Fair	Medium	+	–	–	+	–	Moderate	1	Dryden et al. (2016)	
Canis familiaris	LOE 1	Fair	Small	–	–	–	–	–	Low	1	Fisara & Guerino (2023)	
Canis familiaris	LOE 1	Fair	Small	+	–	–	–	–	Low	1	Labruna et al. (2023)	
Canis familiaris	LOE 1	Fair	Small	–	–	–	–	?	Low	1	Ohmes et al. (2015)	
Canis familiaris	LOE 1	Fair	Small	–	–	–	–	–	Low	1	Zewe et al. (2017)	
Canis familiaris	LOE 1	Fair	Small	–	–	–	–	–	Low	1	Ranjan, Young & Sun (2018)	
Canis familiaris	LOE 1	Fair	Small	–	–	–	+	–	Low	1	Taenzler et al. (2016c)	
Canis familiaris	LOE 1	Fair	Small	–	–	–	–	–	Low	1	Walther et al. (2014b)	
Canis familiaris	LOE 1	Fair	Small	–	–	–	–	–	Low	1	Walther et al. (2014c)	
Canis familiaris	LOE 1	Fair	Small	–	–	–	–	–	Low	1	Allen et al. (2020)	
Canis familiaris	LOE 1	Fair	Small	–	–	–	–	+	Low	1	Allen et al. (2020)	
Canis familiaris	LOE 1	Fair	Small	+	–	–	+	–	Moderate	1	Dryden et al. (2018)	
Canis familiaris	LOE 1	Fair	Very small	–	–	–	–	–	Low	1	Reif et al. (2023)	
Canis familiaris	LOE 1	Fair	Very small	+	–	–	–	–	Low	2	Tahir et al. (2024)	
Canis familiaris	LOE 1	Fair	Very small	–	–	–	?	–	Low	1	Jongejan et al. (2016)	
Canis familiaris	LOE 1	Fair	Very small	–	–	–	?	–	Low	1	Loza et al. (2017)	
Canis familiaris	LOE 1	Fair	Very small	–	–	–	–	–	Low	1	Beugnet, Liebenberg & Halos (2015a)	
Canis familiaris	LOE 1	Fair	Very small	–	–	–	–	–	Low	1	Beugnet, Liebenberg & Halos (2015b)	
Canis familiaris	LOE 1	Fair	Very small	–	–	–	–	–	Low	1	Queiroga et al. (2020)	
Canis familiaris	LOE 1	Fair	Very small	–	–	–	–	–	Low	1	Queiroga et al. (2021)	
Canis familiaris	LOE 1	Fair	Very small	+	–	–	–	–	Low	1	Six et al. (2016a)	
Canis familiaris	LOE 1	Fair	Very small	+	–	–	–	–	Low	1	Six et al. (2016b)	
Canis familiaris	LOE 1	Fair	Very small	–	–	–	+	–	Low	1	Rohdich, Meyer & Guerino (2022b)	
Canis familiaris	LOE 1	Fair	Very small	–	–	–	–	–	Low	1	Taenzler et al. (2015)	
Canis familiaris	LOE 1	Fair	Very small	–	–	–	–	–	Low	1	Taenzler et al. (2016c)	
Canis familiaris	LOE 1	Fair	Very small	–	–	–	–	–	Low	1	Taenzler et al. (2016d)	
Canis familiaris	LOE 1	Fair	Very small	–	+	–	–	–	Low	1	Taenzler et al. (2016a)	
Canis familiaris	LOE 1	Fair	Very small	–	–	–	–	–	Low	1	Taenzler et al. (2017)	
Canis familiaris	LOE 1	Fair	Very small	–	–	–	–	–	Low	1	Toyota et al. (2019)	
Canis familiaris	LOE 1	Fair	Very small	–	–	–	–	–	Low	1	Williams et al. (2015)	
Canis familiaris	LOE 1	Fair	Very small	–	–	–	–	–	Low	1	Burgio, Meyer & Armstrong (2016)	
Canis familiaris	LOE 1	Fair	Very small	?	–	–	–	–	Low	1	Dongus, Meyer & Armstrong (2017)	
Canis familiaris	LOE 1	Fair	Very small	–	–	–	–	–	Low	1	Dryden et al. (2015a)	
Canis familiaris	LOE 1	Fair	Very small	–	–	–	–	–	Low	1	Fourie et al. (2015)	
Canis familiaris	LOE 1	Fair	Very small	–	–	–	–	–	Low	1	Fourie, Meyer & Thomas (2019)	
Canis familiaris	LOE 1	Fair	Very small	–	–	–	–	+	Low	1	Gopinath et al. (2018)	
Canis familiaris	LOE 1	Fair	Very small	–	–	–	–	+	Low	1	Gopinath et al. (2018)	
Canis familiaris	LOE 1	Good	Small	–	–	–	?	?	Low	1	Walther, Allan & Roepke (2015)	
Canis familiaris	LOE 1	Good	Very small	?	–	–	–	–	Low	1	Evans et al. (2023)	
Canis familiaris	LOE 1	Good	Very small	–	–	–	–	–	Low	1	Walther et al. (2014a)	
Canis familiaris	LOE 1	Good	Very small	–	–	–	–	–	Low	1	Walther et al. (2014d)	
Canis familiaris	LOE 1	Good	Very small	–	–	–	–	–	Low	1	Walther et al. (2014e)	
Canis familiaris	LOE 1	Poor	Small	–	–	–	–	–	Low	1	Gomez et al. (2018a)	
Canis familiaris	LOE 1	Poor	Very small	–	–	–	–	–	Low	1	Becskei et al. (2016)	
Canis familiaris	LOE 1	Poor	Very small	–	–	–	–	+	Low	1	Bongiorno et al. (2020)	
Canis familiaris	LOE 1	Poor	Very small	–	–	–	–	–	Low	1	Bongiorno et al. (2022)	
Canis familiaris	LOE 1	Poor	Very small	–	–	–	–	–	Low	1	Gomez et al. (2018a)	
Canis familiaris	LOE 1	Poor	Very small	–	–	–	–	–	Low	1	Gomez et al. (2018b)	
Canis familiaris	LOE 2	Fair	Large	None	–	–	–	–	Low	1	Gurtler et al. (2022)	
Canis familiaris	LOE 2	Fair	Large	None	?	–	?	–	Low/moderate	1	Duangkaew et al. (2018)	
Canis familiaris	LOE 2	Fair	Very small	None	–	–	–	?	Low	1	Busselman et al. (2023)	
Canis familiaris	LOE 2	Fair	Very small	None	–	–	–	?	Low	2	Zineldar et al. (2023)	
Canis familiaris	LOE 2	Fair	Very small	None	–	–	–	–	Low	1	Williams et al. (2015)	
Canis familiaris	LOE 2	Fair	Very small	None	–	–	–	–	Low	1	Williams et al. (2015)	
Canis familiaris	LOE 3	Fair	Medium	–	N/A	–	+	–	Low	1	Crosaz et al. (2016)	
Canis familiaris	LOE 3	Fair	Medium	?	N/A	–	–	–	Low	1	Djuric et al. (2019)	
Canis familiaris	LOE 3	Fair	Medium	+	N/A	–	–	–	Low	1	Fisara et al. (2015)	
Canis familiaris	LOE 3	Fair	Small	+	N/A	–	–	?	Low/moderate	1	Romero et al. (2016)	
Canis familiaris	LOE 3	Fair	Small	+	N/A	–	–	+	Moderate	2	Ortega-Pacheco et al. (2022)	
Canis familiaris	LOE 3	Fair	Very small	+	N/A	–	–	–	Low	2	Hansen-Jones & Ronai (2024)	
Canis familiaris	LOE 3	Fair	Very small	+	N/A	–	–	–	Low	1	Dalmau & Ordeix (2024)	
Canis familiaris	LOE 3	Fair	Very small	+	N/A	–	–	–	Low	2	Morita et al. (2018)	
Canis familiaris	LOE 3	Fair	Very small	+	N/A	–	–	?	Moderate	2	Vargo & Banovic (2021)	
Canis familiaris	LOE 3	Poor	Very small	+	N/A	–	–	–	Low	1	Gaens et al. (2019)	
Canis lupus	LOE 2	Good	Very small	None	–	–	–	–	Low	1	Berny et al. (2024)	
Chrysocyon brachyurus	LOE 3	Fair	Very small	+	N/A	–	–	?	Moderate	2	Fiori et al. (2023)	
Cryptoprocta ferox	LOE 2	Good	Very small	None	–	–	–	–	Low	1	Berny et al. (2024)	
Felis domesticus	LOE 1	Fair	Large	–	–	–	+	–	Low	1	Rohdich et al. (2018)	
Felis domesticus	LOE 1	Fair	Large	–	+	–	+	–	Moderate	1	Meadows, Guerino & Sun (2017b)	
Felis domesticus	LOE 1	Fair	Medium	–	–	–	+	–	Low	1	Dryden et al. (2020)	
Felis domesticus	LOE 1	Fair	Small	–	–	–	–	–	Low	1	Ranjan, Young & Sun (2018)	
Felis domesticus	LOE 1	Fair	Small	–	–	–	–	–	Low	1	Walther, Allan & Roepke (2016)	
Felis domesticus	LOE 1	Fair	Small	–	–	–	–	–	Low	1	Walther, Fisara & Roepke (2018)	
Felis domesticus	LOE 1	Fair	Very small	+	–	–	–	–	Low	1	Vatta et al. (2019a)	
Felis domesticus	LOE 1	Fair	Very small	+	–	–	–	–	Low	1	Vatta et al. (2019b)	
Felis domesticus	LOE 1	Fair	Very small	–	–	–	–	–	Low	1	Taenzler et al. (2017)	
Felis domesticus	LOE 1	Fair	Very small	–	–	–	–	–	Low	1	Taenzler et al. (2018)	
Felis domesticus	LOE 1	Poor	Small	?	–	–	–	–	Low	2	Guimarães et al. (2023)	
Felis domesticus	LOE 2	Fair	very small	None	–	–	–	?	Low	1	Ribeiro Campos et al. (2021)	
Felis domesticus	LOE 2	Poor	Very small	None	–	–	–	?	Low	1	Duangkaew & Hoffman (2018)	
Felis domesticus	LOE 3	Fair	Very small	+	N/A	–	–	–	Low	2	Chuenngam & Chermprapai (2024)	
Felis domesticus (owned)	LOE 1	Fair	Medium	?	–	–	–	–	Low	1	Bosco et al. (2019)	
Felis domesticus (owned)	LOE 1	Fair	Medium	+	–	–	+	–	Moderate	1	Dryden et al. (2018)	
Felis domesticus (owned)	LOE 1	Fair	Small	–	–	–	–	–	Low	1	Fisara, Guerino & Sun (2018)	
Felis domesticus (owned)	LOE 1	Fair	Small	+	–	–	–	–	Low	1	Fisara, Guerino & Sun (2019)	
Felis domesticus (owned)	LOE 1	Fair	Very small	–	–	–	–	–	Low	1	Geurden et al. (2017)	
Felis domesticus (owned)	LOE 3	Fair	Small	+	N/A	–	–	–	Low	1	Briand et al. (2019)	
Felis domesticus (owned)	LOE 3	Good	Very small	+	N/A	–	–	–	Low	2	Bouza-Rapti, Tachmazidou & Farmaki (2022)	
Felis domesticus (stray)	LOE 1	Fair	Small	?	–	–	–	–	Low	1	Bosco et al. (2019)	
Felis domesticus (stray)	LOE 3	Fair	Very small	+	N/A	–	–	–	Low	2	Ilie et al. (2021)	
Lutra lutra	LOE 2	Good	Very small	None	–	–	–	–	Low	1	Berny et al. (2024)	
Mesocricetus auratus	LOE 3	Fair	Very small	+	N/A	–	–	–	Low	2	Brosseau (2020)	
Mus musculus	LOE 2	Good	Large	None	–	–	–	–	Low	1	Pelletier et al. (2024a)	
Nasua nasua	LOE 2	Good	Very small	None	–	–	–	–	Low	1	Berny et al. (2024)	
Oryctolagus cuniculus	LOE 2	Good	Small	None	–	–	–	–	Low	2	Sharaf et al. (2023a)	
Oryctolagus cuniculus	LOE 2	Good	Small	None	–	–	–	?	Low	1	Sharaf et al. (2023b)	
Oryctolagus cuniculus	LOE 3	Fair	Very small	?	N/A	–	–	–	Low	1	Singh et al. (2022)	
Oryctolagus cuniculus	LOE 3	Poor	Small	?	N/A	–	–	–	Low	1	Sheinberg et al. (2017)	
Oryctolagus cuniculus	LOE 3	Poor	Small	?	N/A	–	–	–	Low	2	d’Ovidio & Santoro (2021)	
Panthera leo	LOE 2	Good	Very small	None	–	–	–	–	Low	1	Berny et al. (2024)	
Panthera onca	LOE 2	Good	Very small	None	–	–	–	–	Low	1	Berny et al. (2024)	
Peromyscus maniculatus	LOE 1	Fair	Large	–	–	–	–	?	Low	1	Pelletier et al. (2022)	
Peromyscus maniculatus	LOE 1	Good	Medium	–	–	–	+	–	Low	1	Pelletier et al. (2020)	
Peromyscus maniculatus	LOE 3	Fair	Large	–	N/A	–	–	–	Low	1	Pelletier et al. (2024a)	
Phascolarctos cinereus	LOE 3	Fair	Very small	+	N/A	–	+	?	Moderate/high	2	Young et al. (2024)	
Puma concolor	LOE 2	Good	Very small	None	–	–	–	–	Low	1	Berny et al. (2024)	
Saguinus midas	LOE 2	Good	Very small	None	–	–	–	–	Low	2	Churgin et al. (2018)	
Saguinus midas	LOE 3	Fair	Very small	+	N/A	–	–	–	Low	2	Tokiwa et al. (2024)	
Speothos venaticus	LOE 2	Good	Very small	None	–	–	–	–	Low	1	Berny et al. (2024)	
Suricatta suricatta	LOE 2	Good	Very small	None	–	–	–	–	Low	1	Berny et al. (2024)	
Ursus americanus	LOE 3	Good	Small	+	N/A	–	–	–	Low/moderate	2	Van Wick et al. (2020)	
Ursus arctos arctos	LOE 3	Fair	Very small	+	N/A	–	–	–	Low	2	Oleaga et al. (2024)	
Vicugna pacos	LOE 3	Fair	Very small	+	N/A	–	–	–	Low	2	Sala et al. (2024)	
Vombatus ursinus	LOE 2	Good	Very small	+	–	–	–	+	Moderate	1	Wilkinson et al. (2021)	
Vombatus ursinus	LOE 2	Good	Very small	+	–	–	–	+	Moderate	1	Wilkinson et al. (2021)	
Vombatus ursinus	LOE 3	Fair	Very small	+	N/A	–	–	–	Low	2	Næsborg-Nielsen et al. (2024)	
Vombatus ursinus	LOE 3	Fair	Very small	–	N/A	–	–	+	Low	1	Wilkinson et al. (2021)	
Note:

LOE, Level of Evidence; Q, journal quartile.

FLU was used to treat various ectoparasites (ticks, mites, fleas, lice, sand flies, triatomine bugs, mosquitoes, and human botfly) on 14 mammalian species including dogs, cats, rabbits, mice, golden (Syrian) hamsters (Mesocricetus auratus), cattle (Bos taurus and B. indicus), alpacas (Vicugna pacos), red-handed tamarins (Saguinus midas), African pygmy hedgehog (Atelerix albiventris), raccoon dogs (Nyctereutes procyonoides), maned wolf (Chrysocyon brachyurus), Eurasian brown bear (Ursus arctos arctos), bare-nosed wombats (Vombatus ursinus), and koalas (Phascolarctos cinereus) (Table 2). Articles investigating the pharmacokinetics of FLU have been conducted on dogs, cats, mice, American black bears (Ursus americanus), lions (Panthera leo), pumas (Puma concolor), jaguars (Panthera onca), fossa (Cryptoprocta ferox), meerkats (Suricatta suricatta), Iberian wolves (Canis lupus), bush dog (Speothos venaticus), South American coati (Nasua nasua), Eurasian otter (Lutra lutra), and wombats (Table 3). No large pharmacokinetic studies have been conducted. The only medium-sized studies were for dogs, cats and mice. One study on American black bears was defined as small, with all other studies including only very small sample sizes.

Table 2 Summary of fluralaner treatment on mammalian species, parasite targeted, method, dosage, efficacy and impact.

Species	Treated for	Administration	Dosage	Health assessment	Outcome	Side effects	No. treated	Reference	
Atelerix albiventris	Caparinia tripilis	O	D0 15 mg/kg	Observational only	D14, Dead mites appeared, skin damage improved, erythema and other symptoms disappeared. D21, test was negative. D30, spines were growing.	None reported	1	Romero et al. (2017)	
Bos taurus	Cochliomyia hominivorax	T	2.5 mg/kg	Observational only	Surgical wounds were created and allowed to be naturally infected with C. honinivorax. Wounds were inspected on D0–D7. No larvae developed.	None reported	42	da Costa et al. (2023)	
Bos taurus	Cochliomyia hominivorax	T	2.5 mg/kg	Observational only	Surgical wounds were created and allowed to be nfected with C. honinivorax eggs. Wounds were inspected on D0–D7. by D1 there was a significant reduction inm myiasis and by D3 no myasis was observed	None reported	36	da Costa et al. (2023)	
Bos taurus	Cochliomyia hominivorax	T	2.5 mg/kg D0, D42 and D84	Observational only	D3 99.5%, D7 to D126 100% efficiacy	None reported	15	Gallina et al. (2024)	
Bos taurus	Dermatobia hominis	T	2.5 mg/kg	Observational only	Cattle naturally infected with Dermatobia hominis were treated on D0, and larval nodule counts performed on D3, D7, D14, D21, D28, D35, D42, D49, D56, D63, D70, D77 and D84 post -treatment. Efficacy D1 73.1%, D3 97.7%, D7–D49 no larvae observed, D70 93.7% and D84 79.9%	None reported	12	da Costa et al. (2023)	
Bos taurus	Haematobia irritans	T	2.5 mg/kg	Observational only	Cattle naturally infected with Haematobia irritans were treated on D0, and larval nodule counts performed on D1, D3, D7, and then weekly until D35, and D49 post-treatment. Efficacy >90% D1–D7 to study 1, and D21 in study 2.	None reported	30	da Costa et al. (2023)	
Bos taurus	Rhipicephalus microplus	T	2.5 mg/kg	Observational only	D0 FLU, and cattle infested on D25, D23, D21, D18, D16, D14, D11, D9, D7, D4 and D2. All ticks that detached on D3-1 were counted. D0 cattle were treated and then infested infested with approx. A total of 5,000 larvae and then twice weekly until D90. All detached ticks were counted after washing the cattle daily. Efficiacy D4 98.3%, D5 99.4%, D6–22 >99.9%, D22–D70 >90%	None reported	16	da Costa et al. (2023)	
Bos taurus	Rhipicephalus microplus	T	2.5 mg/kg	Observational only	D0 cattle were treated, and naturally infected with ticks, counts of ticks occurred on D3, D7, D14, D21 and then weekly until D56, D63 and D77. Efficacy >95% at all sites, except D3 <90% at two sites	None reported	100	da Costa et al. (2023)	
Bos taurus	Rhipicephalus microplus	T	2.5 mg/kg D0, D42, D84, D 126, D158, D210	Observational only	Near 0 counts on all animals throughout study	No adverse effects	15	de Aquino et al. (2024)	
Bos taurus	Rhipicephalus microplus	T	2.5 mg/kg when ticks <4 mm were observed on > or = 30% of the animals (D0, D56, D112, D168)	Observational only	Near 0 counts on all animals throughout study	No adverse effects	15	de Aquino et al. (2024)	
Bos taurus	Rhipicephalus microplus	T	2.5 mg/kg D0, D42 and D84	Observational only	D3 99.5%, D7 to D126 100%	None reported	15	Gallina et al. (2024)	
Bos taurus	Rhipicephalus microplus	T	2.5 mg/kg D0, D42 and D84	Observational only	D0 to D126 100%	None reported	30	Zapa et al. (2024)	
Bos taurus indicus	Rhipicephalus microplus	T	1 ml/20 kg	None described	7 h to D249 >95% efficacy, D336 70.2%	None reported	24	Reckziegel et al. (2024)	
Canis familiaris	Aedes aegypti	O	250 mg FLU	Used blood from dogs	24 h post-feeding mosquito survival reduced to 33.2–73.3%, efficacy D14 14.2–91.4%	None reported	6	Evans et al. (2023)	
Canis familiaris	Amblyomma Americanum	O	D0 25–56 mg/kg, D30, 60 placebo	Observational only	D0, live tick counts were significantly lower, but not different with the sarolaner group. The effectiveness at 8h on day 0 was 45.0%. The effectiveness was 53.4 and 61.4% at 12 h on day 0 and day 14. Days 0, 14 and 28 at 24 h, effectiveness was 97.8%, 85.2% 82.4%. At 24 h, sarolaner provided a significantly lower number of live ticks than fluralaner on days 42 to 90.	None reported	8	Six et al. (2016b)	
Canis familiaris	Amblyomma americanum	O	D0 >25 mg/kg 13.4% w/w FLU	Observational only	>94% efficacy at 48 h to D84	None reported	10	Allen et al. (2020)	
Canis familiaris	Amblyomma americanum	O	D0 >25 mg/kg 13.4% w/w FLU	Observational only	>94% efficacy at 48 h to D28, 87.8% at 48 h and 93.5% 72 h on D28, 68.3% 48 h and 78.0% 72 h on D84	None reported	10	Allen et al. (2020)	
Canis familiaris	Cheyletiella spp.	O	1,000 mg FLU	Observation and trichogram	D7 no longer pruritic and continued to improved and skin scale reduced until at least 11 months, D21, 2Mon, no mites observed under microscope	None reported	1	Hansen-Jones & Ronai (2024)	
Canis familiaris	Cheyletiella spp.	O	27 mg/kg	Observation and trichogram	D28 reduced pruritus, no visible mites, D62 lesions resolved	None reported	1	Hansen-Jones & Ronai (2024)	
Canis familiaris	Cheyletiella spp.	O	49 mg/kg	Observation and trichogram	D28 reduced pruritus, no visible mites, D62 lesions resolved	None reported	1	Hansen-Jones & Ronai (2024)	
Canis familiaris	Ctenocephalides canis	O	D0 25–56 mg/kg	Observational only	The effectiveness of the fluramine group was 99.7%, 99.8% and 99.8% respectively; 91.1%, 95.4% and 95.3% of the primary dogs were recorded as a flea-free after three inspections	Vomiting (7.1%)	224	Meadows, Guerino & Sun (2014)	
Canis familiaris	Ctenocephalides canis	T	D0 25–56 mg/kg	Observational only	The effectiveness of the fluramine group was 99.1%, 99.5% and 99.0% respectively (significantly effective); Flea populations reduced by 93.9%, 96.2% and 93.3% respectively	None reported	221	Meadows, Guerino & Sun (2017a)	
Canis familiaris	Ctenocephalides felis	I	D0 15 mg/kg	Clinical observations and observations of the injection site for 96 h post-injection	Efficacy D2 99.7%, and 100% until D367	One dog with mild erythema on D1 post injection	10	Fisara & Guerino (2023)	
Canis familiaris	Ctenocephalides felis	O	D0, in accordance with the European registration labels	Observational only	6 h-effectiveness from 45.1% to 97.8%. At 12 h-from 64.7% to 100%. 24 h–100% except for day 84 (99.6%).	None reported	8	Beugnet, Liebenberg & Halos (2015b)	
Canis familiaris	Ctenocephalides felis	O	D0 25–56 mg/kg, D30, 60 placebo	Observational only	At 8 h, flea counts were significantly lower in both treatments than in the placebo treated dogs, and sarolaner group lower than fluralaner group on days 74 and 90. At 8 h on days 0 and 44, effectiveness was 100%; on the other count days but D29, the effectiveness of the sarolaner group was higher than the fluralaner group. At 12 h, the effectiveness was 100% except on days 59, 74, and 90. At 24 h, the effectiveness maintained at 100%.	None reported	8	Six et al. (2016a)	
Canis familiaris	Ctenocephalides felis	O	D0 25 mg/kg	Observational only	Fluralaner started to work after 1 h with an effectiveness of 8%. Efficacy at W0(80.5–100%), W4(96.8–100%), W8(91.4–100%), and W12(33.5–100%)	None reported	48	Taenzler et al. (2014)	
Canis familiaris	Ctenocephalides felis	O	D0 and D28 oral FLU (25–56 mg/kg)	Observational only	D28, D84 and D168 had flea allergy dermatitis scaores reduced by 89.8%, 98.8% and 99.8% respectively, and pruritis values reduced to 45.2%, 71.2% and 80.8% respectively.	None reported	20	Crosaz et al. (2016)	
Canis familiaris	Ctenocephalides felis	O	D0 26.5–52.4 mg/kg	Observational only	D7 flea counts reduced by 99.7%, W4+ 99.6–100% efficacy	One dog vomited	18	Dryden et al. (2018)	
Canis familiaris	Ctenocephalides felis	O	D0 27.2–56.3 mg /kg FLU	Observational only	Dogs with no fleas D7 76.5%, D14 88.2%, D21 94.1%, one flea on one dog on D28-30, D40-45 97.1%, and D54-60 and D82-86 100%	lethargy (n = 2) and increased pruritis (n = 1), vomiting after eating grass (n = 1), coughing (n = 2) following FLU treatment	34	Dryden et al. (2016)	
Canis familiaris	Ctenocephalides felis	O	D0 25 mg–56/kg FLU	Observational only	D2-122 100% efficacy	None reported	6	Dryden et al. (2015a)	
Canis familiaris	Ctenocephalides felis	O	D0 25 mg/kg minimum	Observational only	W4, W8 and W12 no dermatitis exhibited, except two dogs in W8	None reported	20	Fisara et al. (2015)	
Canis familiaris	Ctenocephalides felis	T	D0 25 mg/kg	Observational only	Two fleas were found on one treated dog on day 1 after treatment and three fleas were found on another dog on day 14. On all other treatment days, no fleas were found and counts were significantly lower than in the control group. Effectiveness was 96.0% on day 1, 94.1% on day 14 and 100% on all other count days.	None reported	10	Ranjan, Young & Sun (2018)	
Canis familiaris	Demodex canis	O	D0 50 mg/kg	Observational only	D18 a small amount of mites were detected. D60 the skin symptoms had healed and no mites were detected.	D3 diffuse non-itchy erythematous papules were observed but they disappeared within a few days.	1	Morita et al. (2018)	
Canis familiaris	Demodex canis	O	250 mg	Observational only	After treatment, dog’s hair loss improved by week 4. Erythema on the mouth and nose disappeared by week 8, and the mite test came back negative.	None reported	1	Vargo & Banovic (2021)	
Canis familiaris	Demodex canis	O	D0 25–56 mg/kg	Observational only	On days 56 and 84, 98% of the dogs achieved parasitological cure. On day 84, 100% of dogs had no detectable live mites. The effectiveness against juvenile-onset demodicosis was 96%. For adult-onset demodicosis, the effectiveness was 100%. By day 84, 94.0% dogs had no significant hair loss. All other clinical signs were also significantly reduced.	None reported	50	Petersen et al. (2020)	
Canis familiaris	Demodex canis	O	D0 25 mg/kg	PCR	On day 0 (prior the treatment), three out of 10 dogs tested positive for mite on at least one skin site. On day 30, two dogs tested positive. On day 90, there were four positive dogs. The results showed no statistical significance between the differences.	None reported	10	Zewe et al. (2017)	
Canis familiaris	Demodex canis	O	D0,28,56 10 mg/kg	Observational only	D28 there was a 99.7% reduction in the number of mites. The effectiveness was 100% on days 56 and 84. By day 28, five dogs had negative scraping results. On days 56 and 84, all seven dogs had zero mite. All dogs were free of scabs on D56 and D84. Over 90% hair regrowth was observed on D28.	None reported	8	Rohdich, Meyer & Guerino (2022b)	
Canis familiaris	Demodex canis	T	D0 25–56 mg/kg	Observational only	On days 56 and 84, 98% of the dogs achieved parasitological cure. On day 84, 98% of dogs had no detectable live mites. The effectiveness against juvenile-onset demodicosis was 100%. For adult-onset demodicosis, the effectiveness was 96.7%. By day 84, 84.0% dogs had no significant hair loss. All other clinical signs were also significantly reduced.	None reported	50	Petersen et al. (2020)	
Canis familiaris	Demodex mites	O	D0 oral FLU (25–56 mg/kg)	Observational, mite counts and mite DNA detection using qPCR	D28 98.6% reduction, D56 100% skin scrapings, D0 to D112 showed a 1,000× reduction on mite DNA using qPCR	None reported	20	Djuric et al. (2019)	
Canis familiaris	Demodex mites	O	D0 25 mg/kg minimum	Observational, mite counts and lesion scores	D28 99.8%, and D56 and 84 100%	None reported	8	Fourie et al. (2015)	
Canis familiaris	Demodex mites	T	D0 25 mg/kg minimum	Observational, mite counts and lesion scores	D28 99.7%, D56 >99%, D84 100%	None reported	8	Fourie, Meyer & Thomas (2019)	
Canis familiaris	Demodex sp. mites	O	D1 25–50 mg/kg	Observational only	The mite reduction rates for 4 months in the adult group were 92.5%, 99.4%, 100% and 100% respectively. Clinical improvement scores were 1.86, 2.57, 2.77, and 2.85. After 1 month of treatment, the mean number of mites decreased from 22.84 to 2.21, with a range from 1–112 to 0–20, and the difference was statistically significant. The mite reduction rates for 4 months in the juvenile-onset group were 98.5%, 100%, 100% and 100% respectively. Clinical improvement scores were 2.14, 2.67, 2.94, and 3. After 1 month of treatment, the mean number of mites decreased from 90.68 to 4.32, with a range from 2–152 to 0–6, and the difference was statistically significant.	None reported	115	Duangkaew et al. (2018)	
Canis familiaris	Dermacentor reticulatus	O	D0, in accordance with the European registration labels	Observational only	63.4% to 99.1% effective	None reported	8	Beugnet, Liebenberg & Halos (2015a)	
Canis familiaris	Dermacentor reticulatus with Babesia canis	O	D0 FLU, according to label recommendations	IFAT, PCR	The effectiveness of fluralaner was 99.2–100%. All tests related to B. canis were negative.	None reported	8	Taenzler et al. (2015)	
Canis familiaris	Dermacentor reticulatus with Babesia canis	T	D0 FLU, according to label recommendations	IFAT, PCR	The effectiveness of fluralaner was 99.3–100%. All tests related to B. canis were negative.	None reported	8	Taenzler et al. (2016d)	
Canis familiaris	Dermacentor variabilis and Amblyomma americanum	O	D0 250–500 mg Bravecto®	Observational only	Efficacy against D. variabilis ranged from 5–68.6% at 6 h and 8.5–100% at 12 h. Efficacy against A. americanum was 4.2–57.3% at 6 h and 33.6–96.8% at 12 h.	None reported	10	Ohmes et al. (2015)	
Canis familiaris	Dipylidium canium infected Ctenocephalides felis	O	D0 25 mg–56/kg FLU	Observational only	Each day from D35 to D113 100% against tapeworm	None reported	8	Gopinath et al. (2018)	
Canis familiaris	Dipylidium canium infected Ctenocephalides felis	T	D0 25 mg–56/kg FLU	Observational only	Each day from D35 to D113 100% against tapeworm	None reported	8	Gopinath et al. (2018)	
Canis familiaris	Ectoparasites	O	25 mg/kg	Observational only	Recovery in mean 3.6 weeks	None reported	4	Zineldar et al. (2023)	
Canis familiaris	Ehrlichia canis	O	D0 25.25–47.62 mg/kg	PCR and IFA	Two dogs positive, the E. canis transmission blocking efficacy was 66.7%, the E. canis protection efficacy was 69.8%	None reported	8	Jongejan et al. (2016)	
Canis familiaris	Fleas	O	D0 25–56 mg/kg	Observational only	FLU for fleas was more effective, ≥99.2% on all count days. The percentage of flea-free households was also higher (>89.57%).	Two dogs had vomiting/diarrhoea and two dogs had loss of appetite	115 households	Rohdich, Roepke & Zschiesche (2014)	
Canis familiaris	Haemaphysalis longicornis	O	D0 10 mg/kg	Observational only	The effectiveness reached 100% on D2 and 30, and was below 80% on D114.	None reported	6	Toyota et al. (2019)	
Canis familiaris	Haemaphysalis longicornis	O	D0 25 mg/kg	Observational only	The effectiveness was 100% on D2 and 30, and then decreased. But consistent >90%	None reported	6	Toyota et al. (2019)	
Canis familiaris	Haemaphysalis longicornis	O	D0 50 mg/kg	Observational only	Longest time to reach 100% effectiveness, decreasing from day 86.	None reported	6	Toyota et al. (2019)	
Canis familiaris	Ixodes holocyclus	O	D0 25 mg/kg minimum	Observational only	D14 to D115 100%, D143 95.7% efficacy	None reported	24	Fisara & Webster (2015)	
Canis familiaris	Ixodes ricinus	O	500 mg	Used blood from dogs not actual dogs	Could not prevent tick engorgement and ultimately prevent Borrelia burgdorferi	None reported	8	Tahir et al. (2024)	
Canis familiaris	Ixodes ricinus	O	D0 25 mg/kg	Observational only	The effectiveness was 89.6%, 97.9%, 100% and 100% at 4, 8, 12 and 24 h after treatment. The effectiveness at 8 h at weeks 4, 8, and 12 were 96.8%, 83.5%, and 45.8%. The 12 and 24 h efficiencies were greater than 98.1% at each week.	None reported	24	Wengenmayer et al. (2014)	
Canis familiaris	Ixodes ricinus and Ctenocephalides felis	Shampooed + T FLU	D0 FLU, according to label recommendations	Observational only	Efficacy against ticks was 99.2–100% and against fleas was 100%.	None reported	8	Taenzler et al. (2016a)	
Canis familiaris	Ixodes ricinus and Ctenocephalides felis	Water-immersion + T FLU	D0 FLU, according to label recommendations	Observational only	Efficacy against ticks was 99.7–100% and against fleas was 100%.	None reported	8	Taenzler et al. (2016a)	
Canis familiaris	Ixodes ricinus and Ixodes scapularis	O	One dose of FLU	Observational only	All ticks died and the average weight was significantly reduced	None reported	1	Williams et al. (2015)	
Canis familiaris	Ixodes ricinus and Ixodes scapularis	O	One dose of FLU	Observational only	All ticks died and the average weight was significantly reduced	None reported	1	Williams et al. (2015)	
Canis familiaris	Ixodes ricinus and Ixodes scapularis	O	One dose of FLU	Observational only	The immediate effectiveness of fluralaner was 100%. At weeks 4, 8 and 12 was 100%, 99.5% and 94.6%. The average weight and hip index of ticks was significantly reduced.	None reported	6	Williams et al. (2015)	
Canis familiaris	Ixodes scapularis	O	25 mg/kg	None described	12 h 99.7%, 24 h 100%, D28 100%	Diarrhea on D12 in one dog	8	Reif et al. (2023)	
Canis familiaris	Ixodes scapularis	O	500 mg	Used blood from dogs not actual dogs	Could not prevent tick engorgemett and ultimately prevent Borrelia burgdorferi	None reported	8	Tahir et al. (2024)	
Canis familiaris	Lutzomyia longipalpis	O	A single dose	Observational only	Mortality was 100% within 5 months, until day 180 was reduced to 72.5%, 62.5% at D10, 61.2% at D240, 50% at D270, 42.5% at D300, 23.7% at D330 and 15% at D360. The effectiveness of fluralaner was maintained at 100% for D150, decreasing to 68.1% by D180 and <50% at D270 post treatment.	None reported	4	Queiroga et al. (2020)	
Canis familiaris	Lymphedema	U	Unknown	Lymphodema	Unknown	None reported	1	Poláková et al. (2023)	
Canis familiaris	Otodectes cynotis	O	D0 25 mg/kg	Observational only	Mites were found in two dogs on day 14. No mites were found in otoscopy on day 28, but one adult mite was found after ear washing, with an efficacy of 99.8%. Cerumen and debris were both improved	None reported	8	Taenzler et al. (2017)	
Canis familiaris	Otodectes cynotis	T	D0 25 mg/kg	Observational only	Mites were found in one dogs on day 14. No mites were found in otoscopy on day 28, but one adult mite was found after ear washing, with an efficacy of 99.8%. Cerumen and debris were both improved	None reported	8	Taenzler et al. (2017)	
Canis familiaris	Phlebotomus papatasi	O	D0 labelled indications	Observational only	D2 55%, D4 77%	None reported	6	Gomez et al. (2018a),	
Canis familiaris	Phlebotomus papatasi	O	D0 labelled indications	Observational only	D14 85%, D32 60%	None reported	10	Gomez et al. (2018a)	
Canis familiaris	Phlebotomus papatasi	O	D0 500 mg	Observational only	D3 100%, D17 93%, D31 94% and D45 75%	None reported	5	Gomez et al. (2018b)	
Canis familiaris	Phlebotomus perniciosus	O	D0 25 mg–56/kg FLU	Observational only	100% efficacy by D1 and D28, and >50% by D84	None reported	6	Bongiorno et al. (2020)	
Canis familiaris	Phlebotomus perniciosus	O	D0 25 mg–56/kg FLU	Observational only	Efficacy on D0 and D28 100%, D56 99.1% and D84 53–57%	None reported	6	Bongiorno et al. (2022)	
Canis familiaris	Phlebotomus perniciosus	O	D0 25 mg–56/kg FLU	Observational only	Efficacy on D0 98.5%, D28 100%, D56 85.9% and D84 0%	None reported	6	Bongiorno et al. (2022)	
Canis familiaris	Preventative	O	1,000 mg FLU	Extensive pathological and clinical examinations	Autoimmune reaction	Pustular dermatitis, lethargy, hyperthermic (autoimmune reaction)	1	Dalmau & Ordeix (2024)	
Canis familiaris	Rhipicephalus sanguineus	O	D0 25.25–47.62 mg/kg	Observational only	Only D30-56 after 12 h tick count significantly lower, speed of kill was 0–55.2%, immediate drop rates was 0–3.8%, anti-attachment efficacy was 0–56.3%	None reported	8	Jongejan et al. (2016)	
Canis familiaris	Rhipicephalus sanguineus	O	D0 >25 mg/kg 13.4% w/w FLU	Observational only	82.9–97.7% at 48 h, >96% at 72 h efficacy against all live nymphs; from 48 h to D84 >95% efficacy for all livefed ticks	None reported	10	Allen et al. (2020)	
Canis familiaris	Rhipicephalus sanguineus	O	D0 >25 mg/kg 13.4% w/w FLU	Observational only	100% efficacy at 72 h D1, 94% D6, 76.8% D28, 86.3% D56, 91.9% D84 and 100% 72 h to D84	None reported	10	Allen et al. (2020)	
Canis familiaris	Rhipicephalus sanguineus	O	D0 25 mg–56/kg FLU	Observational only	Initial treatment >90% efficacy by 12 h, 100% 24 h. After reinfestation, a reduction of 95.5% until D44 and ranged from 42.2–87.5% to D60	None reported	8	Becskei et al. (2016)	
Canis familiaris	Rhipicephalus sanguineus	O	D0 FLU	Observational only	D0 4 h efficacy 60.2%, 8 h 99.6%, and 12, 24 and 48 h 100% efficacy	None reported	8	Burgio, Meyer & Armstrong (2016)	
Canis familiaris	Rhipicephalus sanguineus	O	D0 sarolaner	Observational only	D0 4 h sarolaner 48.2% efficacy, 8 h 94.7%, and 100% efficacy at 12, 24 and 48 h	None reported	8	Burgio, Meyer & Armstrong (2016)	
Canis familiaris	Rhipicephalus sanguineus	O	D0 imidacloprid + permethrin	Observational only	D0 36.9% efficacy at 2 h, and 80.1% at 48 h	None reported	8	Burgio, Meyer & Armstrong (2016)	
Canis familiaris	Rhipicephalus sanguineus	T	D0 25 mg/kg	Observational only	The effectiveness of the six experiments was 91.1–100%, 100%, 100%, 96.7–100%, 96.6–100% and 99.6–100%, respectively.	1 mild erythema and 1 mild wheal development	56	Taenzler et al. (2016b)	
Canis familiaris	Rhipicephalus sanguineus	T	D0 afoxolaner	Observational only	D0 90.8% efficacy at 8 h, 90.8% efficacies respectively, and 100% efficacy at 48 h.	None reported	8	Burgio, Meyer & Armstrong (2016)	
Canis familiaris	Rhipicephalus sanguineus	T FLU and no water immersion	25–56 mg/kg	Observational only	Acaricidal efficacy: 99.3%; Live attached ticks: 0.3 ± 0.5	None reported	8	Dongus, Meyer & Armstrong (2017)	
Canis familiaris	Rhipicephalus sanguineus	T FLU followed by water immersion after 12 h	25–56 mg/kg	Observational only	Acaricidal efficacy: 99.3%; Live attached ticks: 0.3 ± 0.7	None reported	8	Dongus, Meyer & Armstrong (2017)	
Canis familiaris	Rhipicephalus sanguineus	T FLU followed by water immersion after 24 h	25–56 mg/kg	Observational only	Acaricidal efficacy: 100%; Live attached ticks: 0.0 ± 0.0	None reported	8	Dongus, Meyer & Armstrong (2017)	
Canis familiaris	Rhipicephalus sanguineus	Water immersion 1 h before T FLU	25–56 mg/kg	Observational only	Acaricidal efficacy: 99.6%; Live attached ticks: 0.1 ± 0.4	None reported	8	Dongus, Meyer & Armstrong (2017)	
Canis familiaris	Rhipicephalus sanguineus	O	D0, in accordance with the European registration labels	Observational only	65.7% to 100% effective	None reported	8	Beugnet, Liebenberg & Halos (2015a)	
Canis familiaris	Rhipicephalus sanguineus (sensu lato)	I	D0 15 mg/kg	Clinical observations and observations of the injection site for 96 h post-injection	Efficiacy D9 100%, remained >97.6% until D338, 92.6% D367	One dog with mild erythema on D1 post injection	10	Fisara & Guerino (2023)	
Canis familiaris	Rhipicephalus sanguineus (sensu lato)	O	25–65 mg/kg	None noted	Efficacy by D70 and until D84	None reported	12	Labruna et al. (2023)	
Canis familiaris	Rhodnius prolixus	O	W1 500 mg of micronized fluralaner	Observational only	The article lacks data reporting on the percentage of mortality. Immediate post feeding mortality is also not reported.	None reported	15	Ortega-Pacheco et al. (2022)	
Canis familiaris	Safety	O	50–87 mg fluralaner/kg + 2.6–4.4 mg milbemycin oxime/kg + 26–44 mg praziquantel/kg	Observational only	Four mild skin lesions, two severe eye discharge, one sinus arrhythmia on D28, 1 loose feces with normal feces on D7, 1 excess ear wax, 2 dental tartar.	None reported	10	Walther et al. (2014b)	
Canis familiaris	Safety	O	56 mg/kg	Observational only	Two cases of vomiting, one accompanied by other symptoms of gastroenteritis. 15 abnormal feces, eight reduced body condition score, six abnormal hair or skin.	None reported	8	Walther et al. (2014a)	
Canis familiaris	Safety	O	168 mg/kg	Observational only	Two cases of vomiting, one accompanied by other symptoms of gastroenteritis. 15 abnormal feces, eight reduced body condition score, six abnormal hair or skin.	None reported	8	Walther et al. (2014a)	
Canis familiaris	Safety	O	280 mg/kg	Observational only	Two cases of vomiting, one accompanied by other symptoms of gastroenteritis. 15 abnormal feces, eight reduced body condition score, six abnormal hair or skin.	None reported	8	Walther et al. (2014a)	
Canis familiaris	Safety	O	168 mg/kg	Observational and blood collected	Fluralaner can be quantified at each plasma concentration result.	None reported	8	Walther et al. (2014d)	
Canis familiaris	Safety	O	Scalibor™ protectorband and 27–50 mg/kg FLU	Observational only	Safe	None reported	10	Walther et al. (2014c)	
Canis familiaris	Safety	O	D0 28 mg/kg	Observational only	Severe neurological dysfunction within 24 h of treatment	Oral dysphagia, muscle twitching, head and body tremors, myoclonic jerks and generalised ataxia	1	Gaens et al. (2019)	
Canis familiaris	Sarcoptes scabiei	O	D0 25 mg/kg	Observational only	Ten dogs (58.8%) had positive skin scrapings on day 7. On days 14–28, all dogs showed negative skin scrapings. From day 14, lesions level and pruritus scale were both <50%.	None reported	17	Romero et al. (2016)	
Canis familiaris	Sarcoptes scabiei var. canis	O	D0 25 mg/kg	Observational only	The efficiency was 100% after 4 weeks. Four dogs were cured of crusts, one dog was cured of erythematous papules, and three dogs were cured of pruritus. The number of dogs presenting scales increased.	None reported	9	Taenzler et al. (2016c)	
Canis familiaris	Sarcoptes scabiei var. canis	T	D0 25 mg/kg	Observational only	The efficiency was 100% after 4 weeks. No casts, crusts and erythematous papules were observed. Four dogs were cured of pruritus, but the number of dogs presenting scales increased.	None reported	11	Taenzler et al. (2016c)	
Canis familiaris	Sarcoptes scabieii	O	D0 oral FLU (25–56 mg/kg)	Observational only	D28 94.4%, D56 and D80 100%	None reported	54	Chiummo et al. (2020)	
Canis familiaris	Sarcoptes scabieii	T	D0 topical FLU (25–56 mg/kg)	Observational only	D28 95.7%, D56 and D80 100%	None reported	46	Chiummo et al. (2020)	
Canis familiaris	T. gerstaeckeri	U	FLU doses unknown	Used blood from dogs	100% efficacy within 24 h	None reported	3	Busselman et al. (2023)	
Canis familiaris	T. gerstaeckeri	U	FLU + ivermectin doses unknown	Used blood from dogs	100% efficacy within 24 h	None reported	4	Busselman et al. (2023)	
Canis familiaris	Ticks	O	D0 25–56 mg/kg	Observational only	At weeks 2 and 4, FLU for ticks was more effective (99.9%). At week 8, lower than fipronil (99.7% vs. 100%). At week 12, effectiveness was 100% in both treatment groups. In addition, the percentage of tick-free households was higher in fluralaner treatment group (>97.67%).	Two dogs had vomiting/diarrhoea and two dogs had loss of appetite	108	Rohdich, Roepke & Zschiesche (2014)	
Canis familiaris	Triatoma brasiliensis	O	A single dose	Observational only	M1-7, mortality was maintained at 100%. M1-2, all exposed bugs died within 1 day. The effectiveness decreased to 66% at M8, 57% at M9 and was <50% at M10. In addition, the mortality of T. brasiliensis in the treated obese dogs remained at 100% at M8 and M9 while the mortality in the treated normal weight dogs was 49.7% and 30% respectively.	None reported	4	Queiroga et al. (2021)	
Canis familiaris	Triatoma infestans	O	D0 25 mg/kg	Observational only	High exposure, 153 bugs completely engorged. The mortality rate is 100%.	None reported	3	Loza et al. (2017)	
Canis familiaris	Triatoma infestans, Trypanosoma cruzi	O	D0 25 mg–56/kg FLU, and 71 dogs got another dose after 7 months.	Observational only	M1-22, the level of site infestation: 100–18%. Average bug abundance: 5.5–0.5.	None reported	83	Gurtler et al. (2022)	
Canis familiaris	Tunga penetrans	O	10–18 mg/kg	Visual assessment of tungiasis lesions	D7 to D21 >95%, D28 to D42 100% efficacy	One dog hit by car but noted no adverse clinical effects	32	dos Santos et al. (2024)	
Canis familiaris	Tunga penetrans	O	D0 25 mg–56/kg FLU	Observational only	D7 54.8%, D14 96.7%, D21–60 100%, D90 93.6%, D120 77.4% and D150 36.7% efficacy	None reported	31	dos Santos et al. (2022)	
Chrysocyon brachyurus	Sarcoptes scabiei	U	25–65 mg/kg	Observational only	Improved clinical signs via camera traps	None reported	2	Fiori et al. (2023)	
Felis domestica	Demodex gatoi	T	250 mg FLU and fatty acid supplement	Observational only	No mites visible at 1 month check using cellophane tape impression. Skin and hair improved, and no scale present.	None reported	1	Chuenngam & Chermprapai (2024)	
Felis domestica	Lynxacarus radovskyi	T	Unknown	Observational only	Efficacy D7 64.5%, D14 81.8%, D28 97.6%, D42–98 100%	No adverse effects	10	Guimarães et al. (2023)	
Felis domesticus	Ctenocephalides felis	T	D0 40 mg/kg	Observational only	The effectiveness of the fluramine group was 99.8%, 99.9% and 99.9% respectively (significantly effective); Flea populations reduced by 99.8%, 99.9% and 99.10% respectively	None reported	224	Meadows, Guerino & Sun (2017b)	
Felis domesticus	Ctenocephalides felis	T	D0 60.3 mg/kg	Observational only	Reduced the flea population by 98.74% within 7 days, reaching 100% efficacy by week 12. Scores of pruritus in cats decreased by 61.44% within 7 days and 93.77% by week 12. Improvement in skin condition reached 87.77%.	None reported	43	Dryden et al. (2020)	
Felis domesticus	Ctenocephalides felis	T	D0 40 mg/kg	Observational only	The number of fleas was reduced by 99.9% on day 1 and continued > 99% until day 70. On day 84, 2–34 fleas were detected on five cats. On day 90, 8–25 fleas were detected in four cats	None reported	8	Vatta et al. (2019a)	
Felis domesticus	Ctenocephalides felis	T	D0 280 mg/kg	Observational only	Efficacy 100% D28, 56 and 84	None reported	13	Briand et al. (2019)	
Felis domesticus	Ctenocephalides felis	T	D0 40.7–86.3 mg/kg	Observational only	D7 flea counts reduced by 96.6%, D28–30 (n = 25) 80.6%, 12 weeks (n = 31) 100%	None reported	31	Dryden et al. (2018)	
Felis domesticus	Ctenocephalides felis	T	D0 40 mg/kg FLU and 2 mg/kg MOX	Observational only	D2 to D93 100%, except D58 99.7%	None reported	10	Fisara, Guerino & Sun (2019)	
Felis domesticus	Ctenocephalides felis	T	D0 40 mg/kg	Observational only	The effectiveness of F was 100% except on day 1 (96.1%). Flea counts were significantly lower in the treatment group than in the control group, except on days 7 and 14 (N/A).	None reported	10	Ranjan, Young & Sun (2018)	
Felis domesticus	Demodex cati	T	D0 250 mg (55.5 mg/kg)	Observational only	Efficacy 100% W4–W24	None reported	1	Bouza-Rapti, Tachmazidou & Farmaki (2022)	
Felis domesticus	Demodex gatoi	O	112.5 mg per cat	Observational only	All cats were negative for skin scrapings after 1 month; the lesions had subsided; the female cat had a negative fecal flotation result.	None reported	4	Duangkaew & Hoffman (2018)	
Felis domesticus	Dermatobia hominis	T	D1 40–94 mg/kg	Observational only	Larvae from three of the cats were judged dead after 24 h and those from the other two cats died after 48 h.	None reported	5	Ribeiro Campos et al. (2021)	
Felis domesticus	Fleas	T	D0 40–94 mg fluralaner plus 2.0–4.65 mg moxidectin/kg	Observational only	Geometric mean live flea count reduction was at least 98.9%. Arithmetic mean live flea count reduction was greater than 96.6%. At least 93.3% of cats achieved flea-free. Flea dermatitis improved or was clinically cured in 86.7% of cats, with a clinical cure rate of 53.3%.	D0, one pruritus and one salivation and lethargy. One dyspnoea. One small spots of hair loss and eight mild alopecia.	135 households	Rohdich et al. (2018)	
Felis domesticus	Fleas	T	40 mg fluralaner/kg	Observational only	The number of fleas was reduced by 99.2–100% before D75 and to 98.5% on D91	None reported	66	Cvejić et al. (2022)	
Felis domesticus	Ixodes holocyclus	T	D0 40 mg/kg	Observational only	D14 to D84 100% efficacy	None reported	10	Fisara, Guerino & Sun (2018)	
Felis domesticus	Ixodes ricinus	T	D0 40 mg/kg FLU	Observational only	D56, D84 and D91 <90%	Greasy and spiky hair, and white deposits observed up to 91 days post-treatment	8	Geurden et al. (2017)	
Felis domesticus	Ixodes scapularis ticks	T	D0 40 mg/kg	Observational only	The number of ticks was reduced >99% on days 1–70. On day 84, 5–21 ticks were detected on four cats. On day 90, live ticks were found on all the cats, with numbers ranging from 2 to 23.	None reported	8	Vatta et al. (2019b)	
Felis domesticus	Otodectes cynotis	T	D0 40 mg/kg	Observational only	No mites were detected after 14 days and the efficiency was 100%. Cerumen and debris were significantly reduced.	None reported	8	Taenzler et al. (2017)	
Felis domesticus	Otodectes cynotis	T	40 mg fluralaner/kg and 2 mg moxidectin/kg	Observational only	On D14 and 28, no mites were found on otoscopic examination. On D28, no live mites were found after ear washing. Effectiveness was 100%. Cerumen and debris were improved.	None reported	8	Taenzler et al. (2018)	
Felis domesticus	Safety	T	93 mg fluralaner/kg + 6.25 mg emodepsid/kg	Observational only	Two cats presented with erythema at the site of emodepsid-praziquantel placement, and one cat each presented with saliva, vomiting, sporadic coughing, sneezing, and mild skin irritation on the chin.	Same with outcome	10	Walther, Allan & Roepke (2016)	
Felis domesticus	Safety	T	93 mg fluralaner/kg + 4.65 mg moxidectin/kg + 16.7 mg praziquantel/kg	Observational only	Two cats appeared dandruff-like flakes on the fur. Three cats appeared to have unidentified crusts-like substances.	Same with outcome	10	Walther, Fisara & Roepke (2018)	
Felis domesticus	Ticks	T	D0 40–94 mg fluralaner plus 2.0–4.65 mg moxidectin/kg	Observational only	Geometric mean live tick count reduction was at least 97.2%. Arithmetic mean live tick count reduction was greater than 95.7%. At least 92.8% of cats achieved tick-free.	D0, one pruritus and one salivation and lethargy. One dyspnoea. One small spots of hair loss and eight mild alopecia.	152	Rohdich et al. (2018)	
Felis domesticus	Ticks	T	40 mg fluralaner/kg	Observational only	The number of ticks was reduced by 99.1–100% before D75 and to 98.1% on D90. More effective against Dermacentor reticulatus and Rhipicephalus sanguineus	None reported	40	Cvejić et al. (2022)	
Felis domesticus (owned)	Ctenocephalides felis and Otodectes cynotis	T	D0 40 mg/kg	Observational only	Efficacy 100% on D7, D14, D28, D56 and D84	None reported	25	Bosco et al. (2019)	
Felis domesticus (stray)	Ctenocephalides felis and Otodectes cynotis	T	D0 40 mg/kg	Observational only	Efficacy 100% on D7, D14, D28, D56 and D84	None reported	14	Bosco et al. (2019)	
Felis domesticus (stray)	Demodex cati	T	M1, 2, 3, 4 250 mg/animal (weighing between 2.8–6.25 kg)	Observational only	W3 skin abrasions were negative. Alopecia, erythema, erosions, ulcers, crusting, pruritus and self-trauma disappeared.	None reported	1	Ilie et al. (2021)	
Mesocricetus auratus	Demodex aurati and Demodex criceti	O	D0 and D60 25 mg/kg (3.3 mg) FLU	Observational only	Efficacy 100% by day 28, and no reoccurence to day 120	None reported	1	Brosseau (2020)	
Oryctolagus cuniculus	Sarcoptes scabiei var. cuniculi	O	25 mg/kg	D0, 2, 4, 6, 8, 20, 12, 14, 21, 28, 35, 42, 49, 56 post-treatment clinical and parasitological studies	From D4 highly significant clinical score progression reduction, D10 no crust were observed and hair growth presumed, itching stopped D21, D28 full recovery	No adverse effects	10	Sharaf et al. (2023a)	
Oryctolagus cuniculus	Sarcoptes scabiei var. cuniculi	O	25 mg/kg	D0, 2, 4, 6, 8, 20, 12, 14, 21, 28, 35, 42, 49, 56 post-treatment clinical and parasitological studies	Not described	None reported, but suggested further serum lipid profiles required	10	Sharaf et al. (2023b)	
Oryctolagus cuniculus	Psoroptes cuniculi	O	D0 25 mg/kg	Observational only	D4, four ears (4/30) tested positive, D8 one ear. D12, all ears were negative. The amount of otic exudate gradually decreased post treatment. D8 no rabbits has large otic exudate, D12 all rabbits were ‘low’	None reported	15	Sheinberg et al. (2017)	
Oryctolagus cuniculus	Sarcoptes scabiei	O	D0 25 mg/kg	Observational only	D14, the skin scrapings of all rabbits were negative and only one rabbit still had clinical signs (alopecia and erythema). D21, all rabbits’ clinical signs disappeared.	None reported	12	d’Ovidio & Santoro (2021)	
Oryctolagus cuniculus	Sarcoptes scabiei	O	D0 25 mg/kg	Observational only	72 h, clinical signs began to improve. D14, all rabbits were assessed as mildly or moderately infected. D30, clinical lesions disappeared in all rabbits. D45, all rabbits tested negative.	None reported	8	Singh et al. (2022)	
Peromyscus leucopus	Ixodes scapularis	O	min. 10 mg/kg (baits)	Observational, Histopathological, blood biochemistry	Efficacy by D4 88–100%, D 11 74–100%, D18 & D25 74%	None reported	9	Pelletier et al. (2024b)	
Peromyscus maniculatus	Ixodes scapularis	O	D0 50 mg/kg	Observational/Blood collected	D2 The total number of attachments was 92, the mortality was 93%, the treatment effectiveness was 97%, the mean plasma concentration (Cp) was 13,815 ± 11,585 ng/ml. D28 the treatment effectiveness was 3%, the Cp was 579 ± 885 ng/ml. D45 the Cp was 46.7 ± 0.5 ng/ml	Eight mice died or were euthanized during the test.	29	Pelletier et al. (2020)	
Peromyscus maniculatus	Ixodes scapularis	O	D0 12.5 mg/kg	Observational/Blood collected	D2 The total number of attachments was 70, the mortality was 87%, the treatment effectiveness was 94%, the mean plasma concentration (Cp) was 4,594 ± 6,995 ng/ml. D28 the treatment effectiveness was 4%, the Cp was 208 ± 277 ng/ml. D45 the Cp was 52 ± 1 ng/ml	Eight mice died or were euthanized during the test.	29	Pelletier et al. (2020)	
Peromyscus maniculatus	Ixodes scapularis	O	50–100 mg/kg	Observational only	A total of 1,496 baits were placed, 1,424 (95%) were completely eaten, 24 (2%) were partially eaten and 48 (3%) were untouched. The total application density was 14.5–29 mg/1,000 m2. The model demonstrated that the number of larvae feeding on mice was reduced by 68% and 86% when the bait density was 2.1 and 4.4/1,000 m2 respectively. At only 4.4 bait density, the number of nymphs feeding on mice was reduced by 72%.	None reported	312	Pelletier et al. (2022)	
Peromyscus spp.	Ixodes scapularis	O	50–100 mg/kg FLU	Observation only	Reduced tick prevalence on mice, potentially reduced B. burgdorferi infected ticks in the environment	None reported	282	Pelletier et al. (2024a)	
Phascolactos cinereus	Sarcoptes scabiei	T	136.4 g/kg	Severely diseased	Died overnight after treatment, presume due to secondary infections	Animal died–cause not confirmed	1	Young et al. (2024)	
Saguinus midas	Demodex sp.	O	15 mg/kg	Observational, Histopathological, PCR	Observation after 5 months saw reduced itching, scale and hair growth	No adverse effects	2	Tokiwa et al. (2024)	
Saguinus midas	Demodex sp. mites	O	D0, 30–35 mg/kg	Observational only	W6-complete resolution of all plaque-like lesions on the trunk and extremities, while the nodular facial masses had significantly reduced in size. M3-facial masses had almost completely disappeared and the skin scrapes continued to be negative.	None reported	2	Churgin et al. (2018)	
Ursus arctos arctos	Demodex spp.	U	D0 38.6 mg/kg FLU and 1.92 mg/kg MOX, D 27 3 mg/kg sarolaner, D69 41.5 mg/kg FLU	Full clinical assessment plus additional treatments	Released on D231	None reported	1	Oleaga et al. (2024)	
Vicugna pacos	Sarcoptes scabiei	O	5 mg/kg	Extensive clinical assessments	1Mon negative skin scraping, healing of lesions; 2mon resolution of lesions, negative skin scraping. 3mon full recovery	None reported	1	Sala et al. (2024)	
Vombatus ursinus	Sarcoptes scabiei	T	25 mg/kg	Observational only	SM scores for all wombats declined 100% over the 4 weeks, and all wombats reached optimal body condition within 2 weeks. Eventually reached sustained tick-free.	None reported	3	Wilkinson et al. (2021)	
Vombatus ursinus	Sarcoptes scabiei	U	45–85 mg/kg over six treatments	Observation only	Unknown, slow recovery, signs of reinfection and recrudescence	None reported	1	Næsborg-Nielsen et al. (2024)	
Vombatus ursinus	Sarcoptes scabiei	U	D0 85 mg/kg and 2Mon 45 mg/kg	Observation only	Exhibited clear signs of recovery	None reported	1	Næsborg-Nielsen et al. (2024)	

Table 3 Summary of pharmacokinetics studies in mammals.

Species (n)	Dosage (mg/kg)	Method	Cmax (ng/mL)	t1/2 (days)	AUC (0 → t) day*ng/mL	AUC (0→∞) day*ng/mL	Reference	
Canis familiaris (n = 10)	56	O	7,976 ± 4,239	14.27 ± 2.53	NA	175,778 ± 75,122	Walther, Allan & Roepke (2015)	
Canis familiaris (n = 24)	12.5	IV	7,109 ± −908	15 ± 2	87,198 ± 11,835 (D112)	87,779 ± 12,004	Kilp et al. (2016)	
Canis familiaris (n = 6)	12.5	IV	2,144 ± 860	13	29,665 ± 13,858 (D112)	29,922 ± 13,808	Kilp et al. (2014)	
Canis familiaris (n = 6)	12.5	T	3,58 ± 94	17 ± 4	18,933 ± 3,599 (D112)	19,577 ± 3,982	Kilp et al. (2016)	
Canis familiaris (n = 6)	25	IV	3,948 ± 1,734	12 ± 3	46,115 ± 18,932 (D112)	46,416 ± 18,929	Kilp et al. (2014)	
Canis familiaris (n = 6)	25	T	727 ± 191	21 ± 3	41,243 ± 7,467 (D112)	43,375 ± 8,752	Kilp et al. (2016)	
Canis familiaris (n = 6)	50	IV	5,419 ± 2,086	14 ± 1	70,171 ± 26,412 (D112)	70,531 ± 26,529	Kilp et al. (2014)	
Canis familiaris (n = 6)	50	T	1,698 ± 318	17 ± 3	85,852 ± 17,283 (D112)	93,468 ± 20,424	Kilp et al. (2016)	
Canis lupus* (n = 1)	35.1	O	620,000	5	7.218 (D31)	7.26	Berny et al. (2024)	
Cryptoprocta ferox* (n = 1)	48.7	O	370,295,000	3	1,373.277 (D35)	1,373.579	Berny et al. (2024)	
Felis domestica (n = 24)	5	IV	4,302 ± 435	11 ± 1	21,863 ± 2,160 (D112)	22,141 ± 2,105	Kilp et al. (2016)	
Felis domestica (n = 6)	20	T	757 ± 328 (D9)	13 ± 2	22,065 ± 7,996 (D112)	22,276 ± 7,996	Kilp et al. (2016)	
Felis domestica (n = 6)	40	T	1,850 ± 808 (D6)	12 ± 4	48,161 ± 15,294 (D112)	48,400 ± 15,177	Kilp et al. (2016)	
Felis domestica (n = 6)	80	T	2,399 ± 865	12 ± 1	89,193 ± 21,413 (D112)	89,690 ± 21,479	Kilp et al. (2016)	
Lutra lutra* (n = 1)	52.9	O	2,137,000	26	8.998 (D82)	8.998	Berny et al. (2024)	
Mus musculus (n = 34)**	50	O	7,357.7	NA	9,242.7	9,495.5	Pelletier et al. (2024a)	
Nasua nasua* (n = 1)	57.2	O	453,000	2	41.354 (D42)	41.354	Berny et al. (2024)	
Panthera leo* (n = 3)	23.5	O	44,880,000 ± 28,616,000	12 ± 1	232.796 ± 95.614 (D89)	234.203 ± 95.339	Berny et al. (2024)	
Panthera onca* (n = 1)	28.4	O	16,089,000	8	145.337 (D72)	145.752	Berny et al. (2024)	
Peromyscus leucopus (n = 37)**	50	O	10,070.4	NA	25,112	25,470.6	Pelletier et al. (2024a)	
Puma concolor* (n = 3)	30.3	O	12,909,000 ± 4,954,000	6 ± 1	54.158 ± 19.755 (D49)	54.786 ± 19.755	Berny et al. (2024)	
Speothos venaticus* (n = 2)	54.5	O	NA	NA	NA	NA	Berny et al. (2024)	
Suricatta suricatta* (n = 1)	90	O	15,000	NA	105 (D14)	NA	Berny et al. (2024)	
Ursus americanus (n = 10)	22.42–46.99	O	14.55	4.9	NA (D70)	NA	Van Wick et al. (2020)	
Vombatus ursinus (n = 2)	85	T	16.44	166.5	512.8 (D91)	NA	Wilkinson et al. (2021)	
Vombatus ursinus (n = 5)	25	T	6.2	40.1	152.9 (D91)	NA	Wilkinson et al. (2021)	
Notes:

* Cmax (ng/kg dry weight) and rough estimates only due to sampling schedule.

** Force-fed only.

Cmax = maximum plasma concentration.

t1/2 elimination half-life.

AUC (0 → t) area under the concentration vs. time curve from time 0 to the last measurable concentration (given as D days).

AUC (0 → ∞) area under the concentration vs. time curve from time extrapolated to infinity.

IV intravenous.

T topical.

O oral.

NA not available.

Efficacy of FLU used on dogs

Multiple studies have investigated the efficacy of FLU against ticks involving Amblyomma americanum (lone star tick), Rhipicephalus sanguineus (brown dog tick), Ixodes holocyclus (paralysis tick), Dermacentor variabilis (American dog tick), D. reticulatus (ornate cow tick), Haemaphysalis longicornis (Asian longhorned tick), I. ricinus (castor bean tick), and I. scapularis (blacklegged tick) on dogs. Overall, FLU can reduce the number of ticks on dogs. For Ixodes scapularis the speed of kill was rapid (Reif et al., 2023) with efficacy noted from 8 h post-treatment and reaching 99.7% at 12 h and 100% by 24 h post-treatment, and on D28 post-treatment, FLU remained 100% effective at killing ticks after re-infestation 24 h earlier. Likewise for other tick species, the speed of kill was rapid and treatments remained effective for 12 weeks (Allen et al., 2020; Beugnet, Liebenberg & Halos, 2015a; Taenzler et al., 2016b). A study using injectable FLU (15 mg/kg) has also shown high levels of initial efficacy D9 100%, remaining >97.6% until D338 and reducing to 92.6% on D367 (Fisara & Guerino, 2023).

A comparative study assessing the speed of kill in an artificial feeding system using Borrelia burgdorferi-spiked blood obtained from dogs treated with FLU and dinotefuran-permethrin-pyriproxyfen (DPP; Vectra® 3D) found DPP was more rapid in killing (100% efficacy 1 h post-treatment) Ixodes ricinus and I. scapularis (Tahir et al., 2024). FLU efficacy at 12 h post-treatment against I. ricinus was <28% and for I. scapularis <20%, furthermore, female ticks (94 and 55, respectively) subsequently tested positive for B. burgdorferi, hence would fail to prevent B. burgdorferi infections in dogs (Tahir et al., 2024).

In an earlier study, when oral FLU was used as a treatment on Babesia canis infected D. reticulatus, it reportedly prevented Babesia canis infections in dogs (Taenzler et al., 2015, 2016d). Experiments using both oral and topical FLU showed the effectiveness, based on polymerase chain reaction (PCR) and immunofluorescence assay test (IFAT) analysis, ranged from 99.2% to 100% (Taenzler et al., 2015, 2016d). None of the treated dogs exhibited clinical signs associated with B. canis (Chiummo et al., 2023; Taenzler et al., 2015, 2016d). FLU also reportedly had a blocking effect on the spread of Ehrlichia canis; for eight FLU treated dogs, with only two dogs PCR positive and remaining infected after exposing to R. sanguineus. The E. canis transmission blocking efficacy was 66.7% and the E. canis protection efficacy was 69.8% (Jongejan et al., 2016). Whilst reductions were noted in all the studies investigating the effectiveness of FLU as a preventive treatment for these endoparasites, FLU was not 100% effective at preventing tick bites and subsequently did not ensure prevention of endoparasite infections, regardless of the speed of kill.

For oral treatments, the highest efficacy obtained was 90–100% for A. americanum, R. sanguineus, and D. reticulatus on most of the collection days. Although there were a few situations with lower efficacy, it was still over 60% (Allen et al., 2020; Beugnet, Liebenberg & Halos, 2015a). All of the topical FLU trials had efficacy over 95% from day (D) 2 post-treatment, but one was lower at 91.1%, and protection lasted until D86 (Taenzler et al., 2016b). Additionally, the specific efficacy was influenced by duration after infection, with efficacy significantly higher after 12 h than after 6 h (Ohmes et al., 2015).

Moreover, FLU was demonstrated to have a high degree of effectiveness against other species of ticks (Ixodes ricinus, I. scapularis, I. holocyclus, and Haemaphysalis longicornis). On most of the count days, the efficacy was higher than 90% and reached 100% in some cases (Fisara & Webster, 2015; Petersen et al., 2023b; Toyota et al., 2019; Wengenmayer et al., 2014; Williams et al., 2015). Interestingly, the mean tick weight and the mean tick coxal indices collected from treated dogs were significantly lower than those of ticks collected from control dogs (Williams et al., 2015).

Oral FLU treatment can effectively improve clinical skin signs caused by Demodex spp. mites. Hair loss can decrease by week 4, and erythema on the mouth or nose disappeared from week 8 (Morita et al., 2018; Vargo & Banovic, 2021). After D18, mite counts were significantly reduced on one dog (Morita et al., 2018), and after 56 to 60 days no mites were found (Djuric et al., 2019; Morita et al., 2018). The long-term effect of FLU was also supported in the 3-years post-treatment monitoring however, one dog suffered a relapse of demodicosis after 1 year (Hoshino et al., 2021).

Additionally, administering topical or oral FLU had similar efficacy on mites, with 98% of the dogs achieving parasitological cure from D56 (Petersen et al., 2020). Similar results (99.8% efficacy) have been observed when treating dogs for Otodectes cynotis, with no mites found during otoscopy after oral or topical FLU treatment on D28; however, adult mites were found after ear washing even though volumes of cerumen and debris were reduced (Taenzler et al., 2017). Furthermore, the number of mites reduced on juvenile dogs (98.5–100%) was slightly more than on adult dogs (92.5–99.4%) within 2 months post-treatment (Duangkaew et al., 2018).

Although FLU can significantly reduce the number of mites on infected dogs, Zewe et al. (2017) found that FLU was not statistically significant at reducing Demodex populations on the skin of healthy dogs. On D0 (prior to treatment), three out of 10 dogs tested positive for mites on at least one skin site, but two dogs tested positive by D30, and on D90, four dogs remained positive (Zewe et al., 2017). However, Fourie et al. (2015) compared the efficacy of chewable 25 mg/kg FLU (Bravecto®) to topical Advocate® (minimum 10 mg/kg imidacloprid/2.5 mg/kg moxidectin) against Demodex spp. mites and found mite numbers were reduced by 99.8% on D28 and 100% on D56 and D84 using FLU, and 98%, 96.5%, and 94.7% on D28, D56, and D84, respectively, using Advocate®. Whereas another similar trial by Fourie, Meyer & Thomas (2019) found FLU had superior efficacy when compared to Advocate®, whereby the efficacy of topical FLU was 99.7% on D28, >99% on D56 (one mite on one dog) and 100% on D84, whilst the three treatments of Advocate® were 9.8% effective on D28, 45.4% on D56, and 0% on D84. Likewise in comparative experiments with imidacloprid-moxidectin and sarolaner, FLU was suggested to have higher efficacy (Chiummo et al., 2020; Rohdich, Meyer & Guerino, 2022a, 2022b).

FLU has also been used for the treatment of Sarcoptes scabiei on dogs. The fastest treatment observed by Romero et al. (2016) was by D14–28, with all dogs having negative skin scrapings, and reduced lesion and pruritus scale levels under 50% from D14. Moreover, no meaningful differences between routes of administration were observed. The efficacy on D28 was 94.4% and 95.7% for oral and topical FLU treatment groups respectively, and was 100% for all groups on D56 and D84 (Chiummo et al., 2020). Taenzler et al. (2016c) found dogs treated with oral and topical FLU were all mite-negative in skin scrapings at 4 weeks post-treatment and had shown some improvement in clinical signs of sarcoptic mange infections, including pruritus, scales, and crusts.

Three dogs were recently successfully treated for Cheyletiella spp. using oral FLU (1,000 mg, 27 and 49 mg/kg, respectively) and topical hydrocortisone (Hansen-Jones & Ronai, 2024). The first dog was no longer pruritic by D7, by D21 no mites were observed, and skin scale continued to improve until at least 11 months post-treatment (Hansen-Jones & Ronai, 2024). The other two dogs had reduced pruritis and no visible mites by D28 and the lesions had resolved by D62 (Hansen-Jones & Ronai, 2024).

Most of the journal articles focusing on using FLU as a treatment for fleas on dogs investigated the efficacy against Ctenocephalides felis and C. canis. The effectiveness reached 80.5% 1-week post-treatment and remained high (91.4–100%) over 12 weeks (Meadows, Guerino & Sun, 2017a, 2014, 2017b; Ranjan, Young & Sun, 2018; Taenzler et al., 2014), even up to D122 (Dryden et al., 2015a, 2015b); however, one study (Taenzler et al., 2014) found the efficacy was significantly lower by week 12 (33.5%). No dogs exhibited active dermatitis or adverse clinical events after 4 weeks (Crosaz et al., 2016; Fisara et al., 2015). Protection was prolonged (by D168) when a second dose was given on D84 (Crosaz et al., 2016). Additionally, the speed of FLU treatment was rapid with over 45.1% efficacy by 6 h post-treatment, and over 95% for some tested days. After 12 h post-treatment, the efficacy reached 100% (Beugnet, Liebenberg & Halos, 2015b).

Seven articles compared the effectiveness of FLU with other drugs to control fleas on dogs. FLU had a significant higher efficacy than the combination of Comfortis® and amitraz collars, fipronil/(S)-methoprene, and 3-monthly doses of afoxolaner (Meadows, Guerino & Sun, 2014). A total of 91.1%, 95.4% and 95.3% dogs in the FLU treatment group were flea-free after three visits, compared with 44.7%, 88.2%, and 84.4% in the Comfortis®/amitraz collars group, at weeks 4, 8, and 12, respectively (Meadows, Guerino & Sun, 2014). Fleas in the fipronil/(S)-methoprene group only decreased by 58.8%, 75.3% and 80.8% (Meadows, Guerino & Sun, 2014). After comparing FLU and fipronil only, the efficacy of FLU (99.2–99.9%) was slightly more than fipronil (93–97.3%) but the proportion of FLU-treated dogs (89.57–97.39%) that reached flea-free status was significantly higher than fipronil-treated dogs (62.3–81.97%) (Rohdich, Roepke & Zschiesche, 2014). Moreover, FLU treated dogs had no fleas from D54, whilst afoxolaner treated dogs were flea-free on D84 (Dryden et al., 2016). However, FLU and sarolaner treatment were similar in efficacy and flea reduction counts (Dryden et al., 2018; Six et al., 2016a), and the efficacy of FLU was slightly less than sarolaner at 8 and 12 h post re-infestations on D74 and D90 (Six et al., 2016a). A more recent study describes the efficacy of injectable FLU (15 mg/kg) on dogs with cat fleas (Ctenocephalides felis) found by D2 with 99.7% efficacy and subsequently 100% until D367 (Fisara & Guerino, 2023).

FLU was also effective against sand fleas (Tunga penetrans) by administering oral FLU, with 54.8% of dogs flea-free on D7 and 96.7% by D14 (dos Santos et al., 2022). On D21–60 post-treatment, all dogs were reported to have no fleas; however, fleas returned by D90, with 93.6% of dogs remaining flea-free, followed by 77.4% and 36.7% being flea-free on D120 and D150, respectively (dos Santos et al., 2022). An additional study by dos Santos et al. (2024) investigated the efficacy of a single dose of oral FLU (10–18 mg/kg) on sand fleas and evaluated efficacy based on lesion (tungiasis) appearance resulting from infestations. Efficacy was visually assessed once per week for 6 weeks, and found to be >95% by D7, remained >95% to D21, and reached 100% between D28 and D42 (dos Santos et al., 2024). One dog in the treatment group died during the study, being hit by a car, and no adverse clinical effects were reported including systemic allergic responses, diarrhoea, or vomiting (dos Santos et al., 2024).

Of all the searches, only one article investigated the efficacy of FLU treatment of lice (Linognathus setosus) infections on dogs (Kohler-Aanesen et al., 2017). The efficacy was 85.7% on D1, over 90% by D7, and 100% from D28 to D84. In contrast, the efficacy of permethrin was only 67.5% on D1 post-treatment and never reached 100% during the trial (Kohler-Aanesen et al., 2017), demonstrating the potential of FLU to have a higher effectiveness than permethrin for treatment of lice on dogs.

Zineldar et al. (2023) compared several different treatments (FLU, mange cide ointment benzyl benzoate, salicylic acid, sulphur sublimate, phenol and tar; Mega Pharma), injectable doramectin (Dectomax®; Zoetis Inc., Parsippany, New Jersey, U.S) and mitac 10 (AgroEvo®)) on dogs with different ectoparasite infestations and skin lesions. FLU treatment (25 mg/kg) was used as an oral treatment in combination with an amitraz dip (500 ppm) and additional supportive treatments, every 12 weeks for up to 6 months on four dogs. FLU was effective for all isolated ectoparasites, with no re-infestations occurred after the initial dose, and was the most effective treatment trialled for Demodex mite infestations, which was also the ectoparasite that took the longest time to treat (Zineldar et al., 2023). The mean time to recovery after treatment was 3.6 weeks; however, when also given doramectin it resulted in increased efficacy and faster hair growth (Zineldar et al., 2023),

The efficacy of FLU against other parasites on dogs, particularly those that act as vectors for other parasites and pathogens, has included sand flies (Phlebotomus spp. (Bongiorno et al., 2020; Gomez et al., 2018a) and Lutzomyia longipalpis) (Queiroga et al., 2020), triatomine insects (Triatoma infestans (Busselman et al., 2023; Loza et al., 2017; Rokhsar et al., 2023), T. brasiliensis (Queiroga et al., 2021; Rokhsar et al., 2023), (Busselman et al., 2023) T. gerstaeckeri (Busselman et al., 2023) and Rhodnius prolixus), and mosquitoes (Aedes aegypti) (Duncan et al., 2023; Evans et al., 2023). Oral FLU (25–56 mg/kg) can effectively treat sand flies (P. perniciosus), with some deaths of fully engorged sand flies that fed on treated dogs reported on D1, and all sand flies dead by 24 h post-treatment (Bongiorno et al., 2020). On D28, after 6 h post exposure there were significantly more deaths of fully engorged sand flies that had fed on treated dogs compared to those that fed on untreated dogs, and 100% of those fully engorged sand flies that fed on treated dogs had died by 24 h post treatment (Bongiorno et al., 2020). On D84, FLU decreased in effectiveness and took longer to kill sandflies (up to 96 h), but the mortality was still higher than 50% (Bongiorno et al., 2020). Similarly, the efficacy of FLU against P. papatasi was 100% and 98.5% on D1, 100% for two trials on D28, and 99.1% and 85.9% on D56 in two later trials, whilst it was not effective (0%) on D84 in one of the trials (Bongiorno et al., 2022). Gomez et al. (2018a) and (2018b) also found high efficacy of FLU against flies (P. papatasi). Furthermore, the long-term efficacy of FLU against Lutzomyia longipalpis remained at 100% by D150, 68.1% by D180, and subsequently less than 50% on D270 (Queiroga et al., 2020).

Fully engorged T. infestans feeding on FLU treated dogs achieved 100% mortality during treatment, and the transmission of Trypanosoma cruzi was prevented (Loza et al., 2017). Furthermore, dogs treated with FLU to control T. infestans and T. brasiliensis led to a decreased site infestation from 100% to 18% over 1 to 22 months, and average bug abundance from 5.5 to 0.5, after initial treatment (Gurtler et al., 2022). However, regional prevalence is an influential factor in effectiveness of FLU on preventing the transmission of T. cruzi, and modelling predicted that the number of infected insects decreased immediately after a single FLU treatment in the high prevalence areas, whilst the number of infected dogs briefly increased, prior to decreasing and then gradually returned to the initial number (Rokhsar et al., 2023). When FLU was administered annually for 4 or 6 years, there was an overall downward trend in the number of infected dogs, but a short rise was observed after each administration. The numbers gradually returned to the initial number after the trial ended (Rokhsar et al., 2023). When the trial was established for 1 or 2 years with FLU administered every 90 days, the number of infected bugs was maintained at a very low level and the number of infected dogs continued to decline except for a brief period after the initial dose, and numbers gradually returned to pre-treatment numbers after the trial ended (Rokhsar et al., 2023). In contrast, the influence of treatment was negative in low prevalence regions with semi-sylvatic transmission, where the number of infected dogs increased post-treatment following the death of infected insects (Rokhsar et al., 2023).

Busselman et al. (2023) investigated the potential efficacy of FLU to reduce T. cruzi infections gained through T. gerstaeckeri bites in dogs. Dogs were treated with FLU (n = 3) or FLU and ivermectin (n = 4), blood samples were taken from dogs on D0, D7, D30, D45 and D90, and the blood was subsequently used in a membrane feeding trial against colony-reared triatomines, killing all triatomines within 24 h of blood meal. Ivermectin had no effect when compared to the efficacy of FLU alone.

FLU can also impact two other triatomine insects (Triatoma brasiliensis and Rhodnius prolixus). The mortality of T. brasiliensis was maintained at 100% over a 7-month period, but the effectiveness of FLU decreased to 66% at 8 months, 57% at 9 months, and was less than 50% at 10 months (Queiroga et al., 2021). Interestingly, the mortality of T. brasiliensis in obese dogs treated remained at 100% at 8 and 9 months, while the mortality in treated dogs of normal weight were 49.7% and 30%, respectively (Queiroga et al., 2021). A trial by Ortega-Pacheco et al. (2022) also suggested FLU is effective in killing R. prolixus for up to 12 weeks, and subsequently with reduced efficacy upto week 16; suggesting it may be a possible option for reducing Chagas disease transmission due to reduced numbers of vectors.

FLU has also performed well in controlling Aedes aegypti fed on FLU treated dogs and subsequently blocked the transmission of canine heartworm (Diroflaria immitis) (Duncan et al., 2023). After D2 post-treatment, all mosquitoes that fed on treated dogs were dead, and overall the efficacy at each time point (n = 6) was over 90%, 77.2% and 17.7% at 6 h on D56 and D84, respectively. Mosquitoes survived over 72 h on D30 and D84; however, no D. immitis were detected in dogs based on PCR assessment (Duncan et al., 2023). Evans et al. (2023) also conducted a study investigating the efficacy of oral FLU treated dogs against Aedes aegypti, by visually assessing mosquito survival after feeding on blood collected once per week for 15 weeks. Mosquito survival was reduced to 33.2–73.3% 24 h post-feeding, and efficacy 14 days post-feeding ranged from 14.2 to 91.4%, and mean number of eggs laid up to 13 weeks post-treatment was significantly reduced (Evans et al., 2023). Despite the reduction in mosquito survival and reproduction (vector for canine heartworm), it was noted that a faster killing time or increased deterrent would be required to prevent heartworm infections as these parasites enter the host via the mosquito mouth parts and subsequently FLU would not prevent heartworm infections (Evans et al., 2023).

Importantly, bathing dogs did not impact FLU efficacy; neither did water nor shampoo baths (Dongus, Meyer & Armstrong, 2017; Taenzler et al., 2016a). After comparing FLU with other parasiticides (sarolaner, imidacloprid + permethrin, and afoxolaner), it was described as relatively faster, had higher effectiveness, and was superior in long-term effectiveness (Becskei et al., 2016; Burgio, Meyer & Armstrong, 2016; Six et al., 2016b, 2017).

Efficacy of FLU used on cats

The efficacy of FLU used to treat cats has been assessed against ticks (I. scapularis, I. holocyclus, Haemaphysalis longicornis (Petersen et al., 2023a)), mites (Demodex spp. (Bouza-Rapti, Tachmazidou & Farmaki, 2022; Chuenngam & Chermprapai, 2024; Duangkaew & Hoffman, 2018; Ilie et al., 2021; Matricoti & Maina, 2017), Lynxacarus radovskyi (Han, Noli & Cena, 2016), and O. cynotis (Bosco et al., 2019; Taenzler et al., 2017, 2018)), cat fleas (Ctenocephalides felis) (Bosco et al., 2019; Briand et al., 2019; Dryden et al., 2020; Meadows, Guerino & Sun, 2017a; Ranjan, Young & Sun, 2018; Rohdich et al., 2018; Vatta et al., 2019a), and flies (Dermatobia hominis) (Ribeiro Campos et al., 2021).

Vatta et al. (2019b) assessed the efficacy of FLU against I. scapularis on cats and found a >99% reduction in numbers from D1 to D70. Ticks were however detected on four cats on D84 (n = 5–21) and on D90 (n = 2–23), and it was suggested that this may have occurred as a result of a FLU dosage <40 mg/kg being administered (Vatta et al., 2019b). A similar trial found efficacy reached 100% and remained effective up to D84 in a similar trial against I. holocyclus (Fisara, Guerino & Sun, 2018).

Bravecto® Plus (40 mg/kg FLU and 2 mg/kg MOX) treatment resulted in a similar efficacy to using only FLU. Both treatments resulted in cats being tick-free and was 100% effective in treating H. longicornis, whilst the efficacy was slightly lower in the FLU group from D90 (Petersen et al., 2023a). In another trial, the live tick count reduction was greater than 97.2%, and at least 92.8% of cats were tick-free during four follow-up visits (Rohdich et al., 2018).

Yet, FLU was not as effective when comparing to a combination of selamectin and sarolaner, with selamectin/sarolaner having a longer duration of activity in two trials (Vatta et al., 2019b). The efficacy against I. scapularis resulted in >99% efficacy and lasted to D90, and no cats were recorded as having more than two ticks (>21 ticks in FLU group). Geurden et al. (2017) also indicated a higher efficacy (>95.8%) and longer duration of protection (>91 days) in selamectin/sarolaner group used to treat I. ricinus, whilst the efficacy of FLU was <90% from D56. In addition, the recovery time for the adverse skin reaction due to tick bites was longer in the FLU group than the selamectin/sarolaner group (Vatta et al., 2019b).

FLU effectively treated generalised demodicosis caused by Demodex mites on cats, and maintained protection for up to 6 months (Duangkaew & Hoffman, 2018). By 1 month, all orally FLU treated cats infected with Demodex gatoi had negative skin scrapings, the lesions subsided in the two infested cats, and faecal flotation results for one adult cat in the study was negative (Duangkaew & Hoffman, 2018). Additionally, a 12-year-old cat with a secondary bacterial infection administered oral FLU no longer had pruritus after 1 month, hair in some areas started to grow, and erythema faded (Matricoti & Maina, 2017). By 2 months, the hair regrew completely and skin scrapings were negative (Matricoti & Maina, 2017).

Topical FLU treatment of cats also reached 100% efficacy when treating for Demodex cati, within 1 month, and protected cats for a further 6 months, as long as an additional dose was applied at week 12 (Bouza-Rapti, Tachmazidou & Farmaki, 2022). Ilie et al. (2021) tested the efficacy of Bravecto Plus® and found it worked faster than FLU. Furthermore, there was a notable reduction in skin abrasion and substantial improvements in the clinical condition of cats observed within 3 weeks, including the disappearance of alopecia, erythema, erosions, ulcers, crusting, pruritus, and self-inflicted trauma (Ilie et al., 2021). Chuenngam & Chermprapai (2024) noted a cat had no visible mites 1 month post topical FLU treatment (250 mg) using cellophane tape impression, and the hair and skin had improved, with the skin no longer having any scale.

FLU can also kill L. radovskyi and O. cynotis on cats. Topical FLU has been shown to be effective against L. radovskyi, with efficacy reported on D7 64.5%, D14 81.8%, D28 97.6%, and D42–98 100%; however, the dose was not stated (Guimarães et al., 2023). In an earlier study, L. radovskyi was eradicated by D28, but protection only remained for 4 weeks when oral FLU (250 mg) was provided, and cats were exposed to other infected cats (Han, Noli & Cena, 2016). In studies investigating O. cynotis infestations, mites were not detected from D7 and FLU remained effective up to D84 (Bosco et al., 2019; Taenzler et al., 2017). After administering Bravecto® Plus, no O. cynotis was found via otoscopic examination on D14 and D28, and no live mites were found after ear washing from D28 (Taenzler et al., 2018). Furthermore, the volume of cerumen and debris within the ears was reduced (Taenzler et al., 2018).

Topical FLU was effective at killing fleas on cats, taking effect as early as D7 and remaining effective to D84 (Ranjan, Young & Sun, 2018). In addition, allergic dermatitis and pruritus improved post-treatment and was almost resolved by D84 (Briand et al., 2019). Similarly, Bravecto® Plus showed rapid and remarkable flea-killing capabilities against Ctenocephalides felis (Rohdich et al., 2018), with the flea count reducing by >98.9%. Long-term protection and clinical improvement was confirmed with at least 93.3% of the cats maintaining flea-free during four follow-up visits and allergic dermatitis improvement in 86.7% of cats (Rohdich et al., 2018).

When comparing the efficacy of FLU and Bravecto® Plus to fipronil/(S)-methoprene and selamectin/sarolaner against fleas on cats, the former two treatments had greater efficacy (Meadows, Guerino & Sun, 2017b). In trials with a very large sample size, only 10 to 38.2% of fleas were reduced using fipronil/(S)-methoprene treatments on cats, whilst a reduction of >90% was observed on FLU-treated cats (Meadows, Guerino & Sun, 2017b). Subsequently, another trial determined no fleas were alive after Bravecto® Plus treatment, except on D58 (n = 3), however, a mean of 24.2–54 live fleas was found on fipronil/(S)-methoprene-treated cats on each count day (Fisara, Guerino & Sun, 2019). In contrast, Dryden et al. (2020) and Vatta et al. (2019a) both found the effectiveness of FLU to kill cat fleas did not meaningfully differ from selamectin/sarolaner, although the FLU-treated cats showed pruritus improvement (Dryden et al., 2020). When FLU was only compared to selamectin, FLU had significantly higher efficacy with >95% from D7 compared to 79.4% on D7 and >90% by D40 for the selamectin-treated group (Dryden et al., 2018).

Another article determined tapeworm (Dipylidium canium) infections of cats, transmitted by fleas, can be completely prevented for up to 12 weeks after treatment with FLU (Gopinath et al., 2018).

Topical FLU has also been shown to be effective at reducing furuncular myiasis caused by Dermatobia hominis (Ribeiro Campos et al., 2021). A female cat with two wounds and one male cat with a wound at the base of the tail were administered 48 mg/kg FLU, whilst a second and a fourth male cat with wounded tails were treated with 61 and 65.8 mg/kg FLU respectively, and a third male cat with a wound on the right posterior limb, was treated with 52.1 mg/kg FLU (Ribeiro Campos et al., 2021). Larvae found on three of the cats died 24 h post-treatment and those found on the other two cats died after 48 h. However, most of the dead larvae required physical removal which caused some pain (Ribeiro Campos et al., 2021).

Efficacy of FLU used on rabbits

FLU has only been trialled against mites in rabbits, specifically Sarcoptes scabiei (d’Ovidio & Santoro, 2021, 2023; Sharaf et al., 2023a, 2023b; Singh et al., 2022) and Psoroptes cuniculi (rabbit ear mite) (Sheinberg et al., 2017). d’Ovidio & Santoro (2021) found 25 mg/kg oral FLU was effective at treating Sarcoptes scabiei infections on rabbits, with all skin scrapings negative on D14 post-treatment, and all clinical signs disappearing by D21, with protection remaining for >90 days. Bravecto® Plus was similarly found to be as effective as FLU at killing mites on rabbits, with skin scrapings and clinical signs also negative for mites after D14 and D21, respectively, and protection lasting up to D90 (d’Ovidio & Santoro, 2023). Sharaf et al. (2023a) likewise found 25 mg/kg of FLU to be effective at treating rabbits for Sarcoptes scabiei with clinical improvements evident from D4 post-treatment, crusts starting to fall off but increased pruritis. By D10 no crusts remained and new hair was beginning to grow, itching ceased by D21 and complete recovery occurred by D28 (Sharaf et al., 2023a). Furthermore, Sharaf et al. (2023b) investigated the immunological and biochemical effects of FLU treatment on rabbit organs after one treatment but noted this differed to studies conducted by the European Medicine Agency (2024) and Wilkinson et al. (2021) that found reduced total cholesterol and recommended further studies on serum lipid profiles. However, Singh et al. (2022) found that it took up to D45 before rabbits were mite-free using FLU, despite clinical skin lesions improving rapidly by 72 h post-treatment and disappearing by D30. Finally, Sheinberg et al. (2017) assessed the efficacy of one 25 mg/kg oral FLU against Psoroptes cuniculi and found four rabbits (4/30) tested positive for mites on D4 and one rabbit by D8, with all rabbits free of mites from D12. The volume of otic exudate also gradually decreased post-treatment, reaching a ‘low’ otic exudate condition on D12 (Sheinberg et al., 2017).

Efficacy of FLU used on rodents

Few FLU efficacy studies have been conducted on rodents. Brosseau (2020) treated a golden (Syrian) hamster infected with Demodex aurati and a few Demodex criceti using oral canine FLU (3.3 mg; 25 mg/kg) on D0 and D60. The skin scrapings indicated the hamster was mite-free on D30, hair re-growth was complete on D60, and no further hair loss was observed on D90 and D120 (Brosseau, 2020).

The efficacy of FLU against I. scapularis on Eastern deer mice (Peromyscus maniculatus), and subsequently transmission of Lyme disease, has been assessed by Pelletier et al. (2020). Treated mice were orally administered FLU at a dose of 50 or 12.5 mg/kg, and mice infested with 20 unfed tick larvae on D2, D28 and D45. On D2, the mean reduction in the number of attached larvae in both treatment groups was significantly higher than in the control group from 12 to 48 h post-infection (Pelletier et al., 2020). The mortality and FLU efficacy on larvae was 93% and 97%, respectively, in the 50 mg/kg treatment group, and in the 12.5 mg/kg group, mortality was 87%, and efficacy 94% (Pelletier et al., 2020). However, the efficacy of the 50 and 12.5 mg/kg groups decreased to 3% and 4% by D28 respectively. The difference in mortality among the three groups was not statistically significant by D45 (Pelletier et al., 2020).

Subsequently, Pelletier et al. (2022) reported a similar trial conducted under field conditions on Peromyscus spp., as they are the main host of Borrelia burgdorferi. FLU was mixed in a peanut butter bait at a dose of 4.8 mg/g for each mouse (n = 312). Each bait station was loaded with 250–500 mg of bait (1.2–2.4 mg FLU), equating for a mouse weighing approximately 25 g receiving a dose of 50–100 mg/kg of FLU (Pelletier et al., 2022). Of all 1,496 baits, 1,424 (95%) were completely eaten. The total application density was 14.5–29 mg/1,000 m2. The model demonstrated that the number of larvae feeding on mice reduced by 68% and 86% when the bait density was 2.1 and 4.4/1,000 m2 respectively. Nevertheless, the number of nymphs feeding on mice reduced by 72% at only 4.4 baits/1,000 m2 density (Pelletier et al., 2022). Pelletier et al. (2024a) subsequently conducted similar field trials using FLU containing baits (50–100 mg/kg) on P. leucopus exposed to Ixodes scapularis, the vector for B. burgdorferi, and found that doses as low as 10 mg/ml could prevent infestation; however, it was required every 7 days. Furthermore, the study resulted in reduced prevalence of ticks on mice and potentially reduced the density of B. burgdorferi-infected ticks in the environment (Pelletier et al., 2024a).

Efficacy of FLU used on cattle

Recently four studies have described the efficacy of topical FLU against the cattle ticks (Rhipicephalus microplus) on cattle (da Costa et al., 2023; de Aquino et al., 2024; Reckziegel et al., 2024). da Costa et al. (2023) treated cattle in Brazil with 2.5 mg/kg FLU and found it to be effective at controlling cattle ticks that were resistant to amitraz (250 ppm), abamectin and ivermectin (500 ug/kg) pour-on, injectable ivermectin (200 and 630 ug/kg), pour-on fipronil (1 mg/mg), injectable fluazuron (1.6 mg/kg) plus ivermectin (630 ug/kg), and diflubenzuron.

Likewise, de Aquino et al. (2024) trialled pour-on FLU efficacy against cattle ticks using two regimens. The first regimen treated cattle every 42 days, and the second when tick larvae <4 mm were observed on cattle, resulting in the cattle in the first regime being treated six times, and those in the second regimen being treated four times, throughout the year long trial (de Aquino et al., 2024). In both trials, de Aquino et al. (2024) found after the initial treatment that tick numbers were reduced to near zero on all individuals; however, between D259 and D343 the tick numbers did not differ to the control cattle (no treatment), presumably due to reduced tick load in the paddock due to lower temperatures (winter). de Aquino et al. (2024) suggested using rotation treatments may reduce the occurrence of resistance, as observed in other studies, using up to six treatments per year. Subsequently it may be more effective to treat cattle for cattle ticks based on observation of ticks (four times in this study), rather than at set times (six times in this study) (de Aquino et al., 2024).

Another study by Reckziegel et al. (2024) compared the impact of one treatment of pour-on 5% FLU (1 ml/20 kg) at the beginning of the rainy season (and three further treatments on D182, D224 and D266), to one treatment applied at the beginning and another at the end of the rainy season against cattle ticks on cattle in Brazil. FLU efficacy was >95% from D3 to D294, except on D40 where it was 94.2%, and on D336 efficacy was 70.2% (Reckziegel et al., 2024).

A study by Zapa et al. (2024) investigated the efficacy of 2.5 mg/kg pour-on FLU (D0, D49, and D70, followed by retreatments on D28, D42, and D42) on cattle ticks on cattle, and subsequently its impact on cattle tick fever, caused by Anaplasma marginale, Babesia bovis, and B. bigemina. In addition to observation of ticks on cattle, blood smears, PCRs, and indirect enzyme-linked immunosorbent assays were used to identify tick fever parasites (Zapa et al., 2024). The study found cattle treated with FLU did not affect Anaplasma marginale and B. bigemina infected cattle negatively (Zapa et al., 2024), and that enzootic stability was reached by 6 to 8 months (Zapa et al., 2024).

In a further study investigating the efficacy of FLU on cattle ticks, and subsequently myiasis, Gallina et al. (2024) treated cattle with 2.5 mg/kg of pour-on FLU on D0, D42 and D84, and conducted tick counts on D3, D7, D14, D28, D35, D42, D56, D70, D84, D98, D112 and D126. FLU efficacy was 99.5% on D3 and 100% D7 to D126, with no Cochliomyia hominivorax (New World screwworm) infections observed.

da Costa et al. (2023) also trialled FLU against C. hominivorax larvae, Dematobia hominis (cattle grub) larvae and Haematobia irritans (horn fly) in cattle, as well as the efficacy of pour-on FLU against Cochliomyia hominivorax (New World screwworm) larvae, Dematobia hominis (cattle grub) larvae and Haematobia irritans (horn fly), ectoparasites that are also highly resistant to many treatments. FLU was effective at preventing and treating myiasis caused by Cochliomyia hominivorax, was 97.7% effective within three days post-treatment in cattle with Dematobia hominis infections and remained >90% effective for up to 70 days post-treatment (da Costa et al., 2023). FLU also showed effectiveness against Haematobia irritans infestations, however, it was highly variable between sites with an average >90% between days 7 to 21 post-treatment.

Efficacy of FLU used on other mammals

Only a few studies have assessed the efficacy on other mammal species, and most have assessed the efficacy against mites (Churgin et al., 2018; Fiori et al., 2023; Hyun et al., 2019; Næsborg-Nielsen et al., 2024; Romero et al., 2017; Sala et al., 2024; Sharaf et al., 2023a, 2023b; Tokiwa et al., 2024; Young et al., 2024) and ticks (da Costa et al., 2023; Wilkinson et al., 2021).

Sala et al. (2024) described no adverse effects after the use of oral FLU (5 mg/kg) on an alpaca with sarcoptic mange. One month after treatment the alpaca had negative skin scrapings, and the lesions were healing (Sala et al., 2024). By 2 months the lesions had resolved and by 3 months the animal had completely recovered (Sala et al., 2024).

Churgin et al. (2018) treated intrafollicular Demodex spp. mites on two adult sibling red-handed tamarins using a single dose of oral FLU (30–35 mg/kg). Approximately 6 weeks post-administration, examination under general anaesthesia confirmed complete resolution of all plaque-like lesions on the trunk and extremities while the nodular facial masses had significantly reduced in size (Churgin et al., 2018). The facial masses had almost completely disappeared in a follow-up examination at 3 months, and the skin scrapings remained negative (Churgin et al., 2018).

Tokiwa et al. (2024) more recently described FLU treatment of Demodex midae in two tamarins (Tokiwa et al., 2024). The tamarins were treated with one 15 mg/kg of oral FLU treatment (Tokiwa et al., 2024). No adverse effects were noted and after 5 months itching had reduced, scale had disappeared and hair regrown (Tokiwa et al., 2024).

An African pygmy hedgehog infected with the mite Caparinia tripilis was fed FLU chewable tablets at a dose of 15 mg/kg (Romero et al., 2017). The skin scrapings were performed on D7, D14, D21, D30, D60, D90 and D120, with many adult stage mites observed on D7, and dead from D14 (Romero et al., 2017). The skin damage improved subsequently, with the erythema and other symptoms disappearing (Romero et al., 2017). The hedgehog tested negative for mites from D21, and the spines re-grew from D30 (Romero et al., 2017).

A trial tested the clinical efficacy of FLU against Sarcoptes scabiei on free-ranging raccoon dogs (Hyun et al., 2019). Six raccoon dogs were administrated a single oral FLU tablet with a mean of 29.8 mg/kg (Hyun et al., 2019). The skin scrapings for all raccoon dogs were negative on D7, thereafter the skin lesions significantly improved and each raccoon dog increased in body weight post-treatment (Hyun et al., 2019).

A study by Fiori et al. (2023) investigated sarcoptic mange in 52 endangered maned wolf of which ten were confirmed by biopsies or skin scrapings. Two additional animals suspected to be affected by sarcoptic mange, but had negative skin scrapings despite alopecia, scaling and skin crusting, were treated with FLU (25–65 mg/kg), and showed improved clinical signs observed on camera traps (Fiori et al., 2023).

One Eurasian brown bear cub was treated for demodicosis with Bravecto® Plus (FLU; 38.6 mg/kg) and moxidectin (1.92 mg/kg) on day one, sarolaner (3 mg/kg) on day 27, and FLU (41.5 mg/kg) on day 69 (Oleaga et al., 2024). Throughout treatment the bear was also given antimicrobial, anti-inflammatory, and analgesics therapy due to the severity of the disease. By day 27 the bear had no further inflammation of the skin lesions (Oleaga et al., 2024). By day 69 the skin appeared normal, hair had started to regrow, and no mites were present (Oleaga et al., 2024). By day 231 the hair had fully regrown, and the bear was released (Oleaga et al., 2024).

FLU efficacy has been assessed on three bare-nosed wombats, including one juvenile male, one juvenile female and one adult male (Wilkinson et al., 2021). FLU was administered topically at a dose of 25 mg/kg to treat sarcoptic mange, caused by Sarcoptic scabiei; however, only tick counts were performed, and no mite assessments undertaken (Wilkinson et al., 2021). All wombats in the study were found to be tick-free within a week and remained tick-free for >15 weeks (Wilkinson et al., 2021). All wombats reached optimal body condition within 2 weeks, and a reduction in sarcoptic mange scores occurred by 50%, 18%, and 10%, respectively, in the first week, and reduced by 100% in juveniles and 72% in the adult after 3 weeks. The sarcoptic mange score for the adult had decreased by 100% in week 4 (Wilkinson et al., 2021).

One more recent study in wombats tracked the skin microbiota during in situ treatment for sarcoptic mange using FLU (Næsborg-Nielsen et al., 2024). The first wombat was treated with FLU five or six times (unclear) with 45 or 85 mg/kg, showed slow signs of recovery and relapsed, whilst the second wombat was treated twice with 45 mg/kg and then 85 mg/kg and recovered (Næsborg-Nielsen et al., 2024).

Young et al. (2024) described a retrospective study of koalas with sarcoptic mange. Of the two koalas treated for sarcoptic mange, only one was treated with FLU (136.4 g/kg) and the animal died. It was presumed the koala died as a result of severe secondary infections, however no postmortem was conducted to confirm the cause of death (Young et al., 2024).

Pharmacokinetics

There have been only a few studies conducted on the pharmacokinetics of FLU, those being on dogs (Kilp et al., 2016; Kilp et al., 2014), cats (Kilp et al., 2016), American black bears (Van Wick et al., 2020), lions, pumas, jaguars, fossa, meerkats, wolves, bush dogs, South American coatis, Eurasian otters (Berny et al., 2024), and wombats (Wilkinson et al., 2021) (Table 3). Studies have involved treating mammals intravenously (n = 5), topically (n = 7) and orally (n = 17) with dosage rates from 5 to 80 mg/kg. The t½ ranged from 4.9 to 40.1 days, with most around 11 to 21 days (Wilkinson et al., 2021). A study by Berny et al. (2024) on several carnivorous wildlife species only provided indicative preliminary estimates for Cmax values due to their sample scheduling, and used faeces rather than blood samples, hence values were reported in ng/kg, and thus unable to be compared to species in other studies. Interestingly, wombats, the only marsupial examined, had a Cmax of only 6.2 days and a t½ of 40.1 days (Wilkinson et al., 2021), which differed greatly to values reported for eutherian mammals. The reason for the very large difference in Cmax and t½ were not discussed but may be due to differences in physiology or metabolism of marsupials and/or wombats, and requires further investigation.

Pelletier et al. (2024a) compared Mus musculus (n = 3) and Peromyscus leucopus (n = 3) given 1,000 mg/kg FLU orally and found no adverse histopathology, including visible or microscopic lesions in any filter organs. Likewise, chemistry parameters appeared normal compared to control mice, except for slightly reduced total protein, and one mouse with high blood nitrogen urea (Pelletier et al., 2024a).

One issue raised by Berny et al. (2024) was the issue of elimination of FLU in faeces, in their study, up to 3 months. Berny et al. (2024) noted that previous studies in rats, dogs and chickens found up to 90% of the administered dose was eliminated in faeces unchanged (referenced in European Medicine Agency (2024)), and supported the findings of Diepens et al. (2023), stating that faeces were likely the most important route of environmental contamination and requires mitigation measures. While environmental contamination as a result of fluralaner use is beyond the scope of this review, the ecological implications of fluralaner warrant further investigation.

Safety concerns and adverse events

A limited number of studies have assessed the safety of FLU on vertebrates, with no studies investigating any long-term effects (Table 2). Not unsurprisingly, the largest number of efficacy studies has been conducted on dogs (n = 75), followed by cats (n = 27), with limited studies on other mammal species. Nineteen studies reported adverse events, and five specifically reported there were no adverse events. Furthermore, our study indicates most of the efficacy studies have been conducted on small to very small sample sizes, and subsequently these small number of studies with small sample sizes correlate to only a small number of studies reporting safety concerns or any adverse events.

In trials using oral FLU on dogs, FLU dosage rates ranging from 25 to 56 mg/kg, lead to potential side effects including vomiting (n = 21), decreased appetite (n = 15), diarrhoea (n = 14), lethargy (n = 14), polydipsia (n = 4), flatulence (n = 3), diffuse non-itchy erythematous papules, loss of appetite (n = 2), pruritis (n = 2), and coughing (n = 2) (Table 2) (Dalmau & Ordeix, 2024; Dryden et al., 2018, 2016; Meadows, Guerino & Sun, 2014; Morita et al., 2018; Reif et al., 2023; Rohdich, Roepke & Zschiesche, 2014). No further adverse effects occurred in dogs when using higher doses of FLU (168 and 280 mg/kg), and feeding patterns were not affected in either study (Walther et al., 2014a, 2014d). In addition, two studies have confirmed the safety of the combination of oral FLU with commercially available milbemycin oxime plus praziquantel or Scalibor™ protector band (deltamethrin) (Walther et al., 2014b, 2014c), and reported no adverse clinical signs related to the combination treatment. In trials using a dose of 25 mg/kg topical FLU, one dog was observed with mild erythema, another with mild wheal development (Taenzler et al., 2016b), and two dogs presented with higher rectal temperatures after being treated with FLU to control ticks (Taenzler et al., 2015, 2016d). One further study investigating a dog with lymphedema stated it had no treatments or vaccinations prior to presentation, with the exception of FLU, and stated the treatment resulted in no clinical signs (Poláková et al., 2023).

One trial has reported signs of severe neurological toxicity 24 h after treatment with 28 mg/kg FLU in a dog (Gaens et al., 2019). Initially the dog exhibited difficultly walking and balancing, which subsequently became worse within an hour (Gaens et al., 2019). The signs related to toxicity included oral dysphagia, muscle twitching, head and body tremors, myoclonic jerks and generalised ataxia; however, the dog did fully recover after 10 h (Gaens et al., 2019). It was noted that the time of neurological dysfunction onset coincided exactly with the expected Cmax time for FLU and suggested a direct relationship between treatment with FLU and neurological dysfunction; however, no definitive identification was able to be determined for the neurological dysfunction (Gaens et al., 2019). Speculation regarding the resultant neurological signs may have occurred due to a gene mutation (MDR1), a combination of individual factors leading to enhanced FLU brain penetration, the result of the dog having been treated with milbemycin oxime plus praziquantel 1 month prior to the FLU being administered, or an alternate reason (Gaens et al., 2019).

Dalmau & Ordeix (2024) described pemphigus foliaceus (cutaneous autoimmune disease) in a dog resulting from the provision of oral FLU (1,000 mg chewable tablet) as a preventative ectoparasite treatment. The dog developed pustular dermatitis and was lethargic and hyperthermic seven days after FLU treatment (Dalmau & Ordeix, 2024). After extensive clinical and pathological investigations, and subsequent pemphigus foliaceus diagnosis, prednisolone therapy was required for 16 weeks to resolve the lesions; presumably as a result of the systematic distribution of FLU and slow plasma elimination (Dalmau & Ordeix, 2024). No recurrence occurred after therapy was withdrawn (Dalmau & Ordeix, 2024).

Four studies have specifically addressed safety and three have reported adverse events in FLU trials on cats (Geurden et al., 2017; Rohdich et al., 2018; Walther, Allan & Roepke, 2016; Walther, Fisara & Roepke, 2018). The fur of eight cats appeared greasy, spiky, and had white deposits after using 40 mg/kg topical FLU (Geurden et al., 2017). A further trial using Bravecto® Plus spot-on solution to control ticks by Rohdich et al. (2018), resulted in one case of pruritus, another exhibiting drooling/lethargy, one having dyspnoea, another cat with small spots exhibiting hair loss, and eight exhibiting mild alopecia. When FLU and emodepsid-praziquantel were administered combined, two cats presented with erythema at the site of emodepsid-praziquantel placement, whilst one cat exhibited drooling, one vomiting, one sporadic coughing, one sneezing, and one mild skin irritation on the chin (Walther, Allan & Roepke, 2016). In contrast, no evidence described any side effects during the trial of the combination of Bravecto® Plus and praziquantel (Walther, Fisara & Roepke, 2018).

Only one article has reported side effects of the spot-on combination of FLU (25 mg/kg) and moxidectin (1.24 mg/kg) on rabbits (d’Ovidio & Santoro, 2023). The rabbit presented with a mild pruritus and erythema approximately 2 h post-treatment which rapidly disappeared within 24 h (d’Ovidio & Santoro, 2023). Additionally, a female raccoon dog received a higher dosage of 52.3 mg/kg oral FLU and subsequently presented with mild diarrhoea post-treatment (Hyun et al., 2019). Whilst Van Wick et al. (2020) found no clinically relevant changes in complete blood counts for bear cubs treated with FLU.

Conclusion

Our study assessed the efficiacy of FLU in clinical trials published in 121 peer-reviewed journal articles on 14 mammalian species and pharmacokinetic studies conducted on 15. Overall, FLU was effective at treating a range of ectoparasites over a short time frame and when there was little chance of re-infection; such as fleas on domestic cats and dogs and other species living in specialised captive environments (e.g. zoological parks). In some cases FLU was effective at reducing the impact of pathogens when the ectoparasites being controlled were reduced; however, it failed to prevent bites from blood-sucking ectoparasites (such as ticks, mosquitoes, and triatomine bugs) and subsequently can not prevent infections from blood-borne pathogens for which those ectoparasites are reservoirs. FLU was deemed moderately safe; however, most trials conducted to date are limited by small or very small samples sizes, and the quality of most studies was ranked as fair, hence is concerning. Furthermore, only a very small number of studies reported on the safety of FLU; and of those studies, all were short-term studies limited to the trial period. Further studies are needed to assess the impacts of FLU on larger numbers of animals, and particularly to identify if there are any impacts on the use of FLU on a larger number of non-model species, as its use on wildlife is increasing. Furthermore, the long-term impacts of using FLU have not been assessed on any species and clearly an investigation into environmental contamination and ecological concerns is warranted.

Supplemental Information

Supplemental Information 1 PRISMA checklist.

Supplemental Information 2 Intended audience.

We would like to thank Dr Hayley Stannard and Amanda Cox for feedback on an earlier version of the manuscript.

Additional Information and Declarations

Competing Interests

The authors declare that they have no competing interests.

Author Contributions

Yuanting Jiang performed the experiments, analyzed the data, prepared figures and/or tables, authored or reviewed drafts of the article, and approved the final draft.

Julie M. Old conceived and designed the experiments, performed the experiments, analyzed the data, prepared figures and/or tables, authored or reviewed drafts of the article, and approved the final draft.

Data Availability

The following information was supplied regarding data availability:

This is a systematic review/meta-analysis.

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
