# Peer review of "A systematic review of fluralaner as a treatment for ectoparasitic infections in mammalian species"

_PeerJ, doi:10.7717/peerj.18882_

## Round 0.1 · original submission · Major Revisions

Please address the concerns raised by the two reviews

·

Basic reporting

First of all, thank you for allowing us to review this scientific reviews.
In general, this review is important in knowing the effects of these parasites on mammals and the progress in treating them.
But I have some comments on that review.
Abstract
Please specify the study hypotheses more clearly in this section and also in the introduction.
It is not recommended to repeat a part that was mentioned in the material part.
Please improve the language more clearly and review the linguistic review of the manuscript as a whole.
Introduction
Please update the references further in the introduction section.
Please add a paragraph about external parasites in this section.
Unification of scientific name at whole manuscript.
Materials & Methods need to be brief and more precise
Results & Discussion: This section in particular needs modifications to be more concise and clear for discussion of the results.
Conclusion :Needs to be improved better

Experimental design

Pretty much acceptable

Validity of the findings

Acceptable

Reviewer 2 ·

Basic reporting

This review provides a thorough examination of how ectoparasites affect various animal groups, including livestock, wildlife, and pets, while also exploring the application of fluralaner (FLU) as an ectoparasiticide. Nevertheless, there are several areas that could be enhanced to bolster the argument, increase precision, and enhance readability. The following critique outlines key suggestions:

While the abstract is well-structured and comprehensive, it could be improved by offering more specific details on efficacy results across different species and clearer insights into pharmacokinetic findings. Additionally, including more information on the methods used to assess study quality would provide a better overview of the reliability across studies.

Introduction: The text effectively illustrates the wide-ranging impact of ectoparasites on various host species. However, it could benefit from a more organized structure. Consider dividing the ecological and economic effects on livestock, pets, and wildlife into separate sections to improve readability and flow.
 Host and parasite examples: The provided examples of different host-parasite relationships are useful, but some sections vary in their level of detail. For instance, while the text discusses ectoparasites of hedgehogs in Libya and wild animals in Madagascar, further explanation of the importance of these specific examples might be necessary. Alternatively, focusing on examples directly relevant to FLU treatment (such as fleas, mites, and ticks in veterinary or wildlife contexts) could help maintain a more focused discussion.
 Evidence of FLU use: It would be advantageous to incorporate additional specific references or studies that elucidate the transition of FLU use from domestic pets to wildlife. A concise delineation of the types of wildlife and the conditions under which FLU is administered would provide context for its expanded application.
 Effectiveness across species: Although there is mention of efficacy variations across species and routes of administration, this point could be further elucidated with examples or hypotheses explaining potential reasons for these inconsistencies. Discussing specific host factors or physiological differences that may contribute to varied FLU efficacy could enhance this section.
 Systematic review justification: At present, the rationale for conducting the systematic review could be articulated more explicitly, particularly given the paucity of comprehensive data on FLU's safety and efficacy across species. Consider explicitly stating gaps in previous research as a foundation for the review.

Experimental design

Database: While "Google Scholar" is noted as the database, it is generally recommended to include multiple databases (e.g., PubMed, Web of Science) to mitigate the risk of omitting key studies, as Google Scholar has limitations. If only Google Scholar was utilized, provide justification for the exclusion of other databases.
 Language and date restrictions: It would be beneficial to clarify if any publication date restrictions were applied, as systematic reviews generally specify the time range.
 Exclusion: The rationale behind excluding descriptive chemical studies and unrelated publications is apparent; however, defining "clearly not relevant" could help standardize exclusion criteria.
 LOE Definition: Clearly defining the Level of Evidence (LOE) classifications with references (e.g., LOE 1 for RCTs, LOE 2 for non-randomized trials, etc.) enhances transparency. Consider briefly describing each level for readers unfamiliar with these categorizations.
 Bias: Utilizing the Cochrane risk of bias tool is appropriate and adds to the review's rigor. Ensure that all Cochrane criteria (selection, performance, detection, attribution, reporting) are clearly defined to avoid ambiguity. Additionally, the basis for "none" and "N/A" ratings for specific biases could be clarified. For example, state why performance bias and selection bias were assigned certain ratings (e.g., "LOE 2 studies automatically received 'none' for selection bias").
 Bias ratings: Defining overall bias ratings, such as "low/moderate" or "moderate/high," would add clarity. Specify how an article was assigned a "moderate" versus a "high" bias rating.
 Enrollment: The enrollment score is a useful addition, but further standardization of "poor," "fair," and "good" criteria could enhance reproducibility. Specifying what constitutes "thorough" reporting of blood collections or clinical examinations might help clarify the "good" rank.
 Ranking: If examples from the studies are available, briefly illustrating why certain studies were rated "poor," "fair," or "good" would make the criteria more concrete.
 Completeness: It is advisable to address how inter-rater reliability was assessed, as systematic reviews typically involve at least two independent reviewers for study selection to mitigate bias. Even if only two authors were involved, reporting agreement rates or incorporating a reliability check would enhance the credibility of the review.
 Quality control for data extraction: Describing the methodology employed for extracting data from the included studies (e.g., independent extraction by authors, verification for consistency) would demonstrate a systematic approach.
 Endnote library: Instead of stating that the "list of 250 references was saved as an Endnote reference library," consider explicating this step in terms of its purpose (e.g., "250 references were imported into Endnote to organize, deduplicate, and categorize studies for systematic screening").

Validity of the findings

Discussion
 Commence with a concise sentence summarizing the overall study distribution (e.g., "This review encompassed 121 studies from 2014-2024 on FLU's effectiveness against ectoparasites in 14 mammalian species").
 The discussion section can be more concise.
Conclusion
 Specify which ectoparasites or types of infections FLU demonstrated the highest efficacy against, and in which cases it proved insufficient. This information will assist readers in understanding the conditions where FLU is likely to be beneficial or inadequate.

---

## Round 0.2 · Minor Revisions

Although the authors have addressed most of the concerns raised by the reviewers, a few minor issues remain that need resolution before acceptance.

·

Basic reporting

Accepted for publication

Experimental design

Accepted

Validity of the findings

Accepted

Additional comments

Accepted for publication

Reviewer 2 ·

Basic reporting

Abstract: The revised abstract is more concise and provides more precise insights into the study's scope, including more details about efficacy and pharmacokinetics.
Introduction: Separating livestock, pets, and wildlife impacts is an good addition. It provides a logical structure and improves readability, making understanding the broader ecological and economic implications easier.
Expanded evidence for FLU: contextual discussion of FLU's application across species is improved, especially with examples from domestic and wildlife settings. This addition aligns well with the systematic review's goals.

Experimental design

Method: The rationale for using Google Scholar is now justified, and the explanation of exclusion criteria (e.g., removing the word "clearly") adds transparency. Additional references for Level of Evidence (LOE) classifications further improve methodological rigour.
The authors have addressed most comments by adding relevant details and clarifications.

Validity of the findings

Pharmacokinetics: the discussion now includes better insights into pharmacokinetics across species, the significant differences in pharmacokinetic parameters, particularly in unique species like wombats, could be further contextualized. For example, what physiological factors might explain these differences?
Bias assessment: Although the Cochrane risk of bias tool was applied, the explanation of ratings such as "none" and "N/A" remains slightly unclear. This section could benefit from examples or a more detailed justification for assigning these ratings.
Discussion of environmental impacts: manuscript mentions concerns about FLU's environmental contamination due to its long detection in feces. Expanding on this topic can add address gaps in the ecological implications of FLU
Conclusion: Although improved, the conclusion remains generic. Emphasizing specific findings (e.g., the most and least practical FLU applications) would make it more impactful and aligned with the study's objectives.

Additional comments

Suggestions:

Language and grammar: improved, some sections (results and discussion) still contain verbose or complex sentences. Further round of linguistic editing can enhance readability.
Recommendation:

The manuscript demonstrates improvements and provides a more organized review of fluralaner's applications and implications. Minor enhancements are needed, the revised version is much closer to publication-ready. It should make a valuable contribution to the field with the suggested refinements.

---

## Round 0.3 · accepted · Accept

The authors have addressed all of the reviewers comments